# Frontier Research on Low-Resource Speech Recognition Technology

**DOI:** 10.3390/s23229096

**Published:** 2023-11-10

**Authors:** Wushour Slam, Yanan Li, Nurmamet Urouvas

**Affiliations:** Xinjiang Laboratory of Multi-Language Information Technology, Xinjiang Multilingual Information Technology Research Center, College of Information Science and Engineering, Xinjiang University, Urumqi 830046, China; 17519525081@163.com (Y.L.); nurmemet@xju.edu.cn (N.U.)

**Keywords:** low-resource speech recognition, deep feature extraction, acoustic models, resource expansion

## Abstract

With the development of continuous speech recognition technology, users have put forward higher requirements in terms of speech recognition accuracy. Low-resource speech recognition, as a typical speech recognition technology under restricted conditions, has become a research hotspot nowadays because of its low recognition rate and great application value. Under the premise of low-resource speech recognition technology, this paper reviews the research status of feature extraction and acoustic models, and conducts research on resource expansion. Especially in terms of the technical challenges faced by this technology, solutions are proposed, and future research directions are prospected.

## 1. Introduction

There are about 6900 languages in the world, fewer than one-tenth of which have more than 1 million speakers [1]. At present, there are only a few languages (such as English and Mandarin Chinese) that have sufficient annotation speech data, and most of the languages face the current situation of lack of data resources, that is, few resources. With the improvement of computer computing power and the perfection of machine learning and other theories, the recognition rate of large-vocabulary continuous-speech-recognition systems has reached more than 90%. Continuous speech recognition is the process of converting continuous speech signals into corresponding text. Usually, speech recognition refers to continuous speech recognition, unless otherwise specified. However, in a low-resource environment, due to the lack of annotated corpus, the model training of the speech-recognition system usually occurs due to overfitting and cannot achieve good recognition performance. As far as current research is concerned, although a large number of targeted studies have made some progress relative to traditional modeling methods, there is still a huge gap between the low-resource speech recognition rate and the large-vocabulary speech recognition rate. According to the literature [2,3,4,5], the word error rate (WER) of a low-resource speech-recognition system based on less than three hours of data is about 50–80%.

In the research of speech recognition in low-resource environments, as early as 2011, the Intelligence Advanced Research Projects Activity (IARPA) proposed the Babel project [6]. The plan lasts for five years, and its research goal is to use limited, multi-recording environment speech data to quickly develop a robust speech-recognition system for any unknown language. Many domestic and foreign research institutions such as Massachusetts Institute of Technology, Cambridge, BBN, IBM, Carnegie Mellon University, Johns Hopkins University, Brno University of Technology, Tsinghua University, and University of Science and Technology of China are participating. With this plan, certain research results have been achieved [7,8,9,10,11,12,13,14]. In addition, research teams such as the University of Edinburgh, Nanyang Technological University, Hong Kong University of Science and Technology, Shanghai Jiao tong University, and Xinjiang University have also conducted a lot of research on speech recognition problems in low-resource environments and have achieved outstanding results. Specifically, the University of Edinburgh has conducted research on cross-language acoustic modeling in the context of Subspace Gaussian Mixture Model (SGMM). Experiments on the GlobalPhone corpus prove that the global parameters of SGMM can be transferred between languages, especially when the parameters are trained in multiple languages. This research provides inspiration for low-resource speech recognition, that is, it is possible to train an acoustic model on low-resource speech data by borrowing SGMM global parameters from one or more source languages and training only state-specific parameters on the target language [15]. The Nanyang Technological University scientific research team proposed a context-dependent phoneme-mapping method by using well-trained models of other languages with a large amount of data and used it for acoustic modeling of low-resource speech recognition. Experiments show that the phoneme-mapping technology is significantly better than the cross-language concatenation method [16]. The Hong Kong University of Science and Technology proposed a solution to the problem of overfitting of the acoustic model using the full covariance matrix training when the training data are insufficient [17]. Shanghai Jiao tong University solved the problem of lack of corpus for low-resource speech recognition by borrowing data from rich languages. The team studied three different data borrowing methods including features, models, and data corpora. Due to the independence between these three strategies, the team believes that they can achieve additive benefits. In order to study the complementarity of the above guesses, the research team combined these three strategies to form an integrated data borrowing framework. Experiments have shown that compared with traditional baselines, the use of an integrated data borrowing framework reduces absolute WER by more than 10% [18]. The Xinjiang University scientific research team conducted research on continuous speech recognition of ethnic languages lacking natural corpus, such as Uyghur, Kazakh, and Kirgiz.

The contributions of this paper are as follows:(1)The research status of low-resource speech recognition is described from three aspects: feature extraction, acoustic model, and resource expansion.(2)The technical challenges faced in realizing low-resource language recognition are analyzed.(3)For low-resource conditions, the future research directions of speech recognition technology are prospected and measures that can be solved are proposed.

## 2. Speech Recognition Technology

Speech is an indispensable and important means for people to convey information, express ideas, and exchange feelings. It is one of the most direct, convenient, and accurate ways of communication between people. Automatic Speech Recognition (ASR), referred to as speech recognition, involves translating continuous speech into corresponding text. Speech-recognition modeling methods are mainly divided into template matching: statistical models and deep models. Next, we will introduce DTW, GMM-HMM, DNN-HMM, and end-to-end models, respectively.

### 2.1. DTW

When the same person speaks the same word, the pronunciation characteristics and duration of the word are often different due to differences in speech speed and intonation, which results in the situation that the speech data obtained by sampling cannot be aligned on the time axis. If the time series cannot be aligned, the traditional Euclidean distance cannot effectively measure the true similarity between the two series. The DTW is proposed to solve this problem. It is an effective method to align two unequal time series and measure the similarity between the two sequences.

As shown in Figure 1, DTW adopts the algorithm idea of dynamic programming, realizes the unequal length matching of two voices P and Q through time bending, and converts the voice matching similarity problem into the optimal path problem. DTW is a typical method in template-matching method that is very suitable for small vocabulary isolated word speech-recognition system. However, DTW relies too much on endpoint detection and is not suitable for continuous speech recognition. DTW has better recognition effect on specific people. Because this article describes low-resource continuous speech recognition, DTW will not be described in the following pages.

### 2.2. GMM-HMM/DNN-HMM

HMM is a statistical analysis model developed on the basis of Markov chains to describe doubly stochastic processes. HMM has the advantages of mature algorithm, high efficiency, and easy training and is widely used in speech recognition. It is still the mainstream technology in speech recognition.

As shown in Figure 2, HMM includes 5 states S1, S2, S3, S4, and S5; each state corresponds to multiple frames of observations, and these observations are feature sequences, increasing along time t, diverse and not limited to taking a range of values, so its probability distribution is not discrete but continuous. Many signals in nature can be represented by Gaussian distributions, including speech signals. Due to the large differences in pronunciation of different people, the specific performance is that the sequence of observations corresponding to each state is diversified, and it is often not enough to describe its distribution simply by a Gaussian function, so more Gaussian combinations are used to characterize more complex distributions. This model that uses GMM as the HMM state to generate the probability density function of the observed value is GMM-HMM. As shown in Figure 2, the GMM corresponding to each state is composed of two Gaussian functions.

A deep neural network (DNN) consists of an input layer, multiple hidden layers, and an output layer. Each layer has fixed nodes, and nodes between adjacent layers are fully connected, as shown in Figure 3. DNN has stronger representation ability, which can model complex speech changes. The GMM of GMM-HMM is replaced by DNN, as shown in Figure 4, and the transition probability and initial state probability of HMM remain unchanged. This subsection focuses on the basic structure of DNN-HMM. 

In addition, DNN has different structures, including convolutional neural network (CNN), long short-term memory network (LSTM), and time-delay neural network (TDNN). One point to emphasize is that the HMM architecture of DNN-HMM is generic, but DNN can use different network models such as CNN, LSTM, and TDNN etc.

Figure 5 is the basic framework of speech recognition corresponding to GMM-HMM/DNN-HMM, including the training process and the recognition process, and consists of four key modules: acoustic model, language model, pronunciation dictionary, and decoder. During the training process, speech data and text data resources are used to train acoustic models and language models, respectively. In the recognition process, the decoder orders the characteristic sequence of the speech signal into the corresponding word sequence, according to the trained acoustic model and language model and pronunciation dictionary.

The purpose of ASR is to recognize a piece of speech as a corresponding word or character sequence, so that the machine can recognize and understand human language. Mathematically describe that by analyzing the observed speech sample sequence O and converting it into the corresponding word sequence W, among $O=\left\{o_{1},o_{2},o_{3},\ldots ,o_{T}\right\}$, $o_{t}$ is the $t_{th}$ frame acoustic feature vector of the speech signal. From a statistical point of view, calculate the optimal word sequence $W^{*}$ to maximize the posterior probability of the word sequence $W^{*}$, as shown in Formula (1):(1)$$W^{*}=\underset{w}{\arg\max}p(W|O)$$

From the Bayesian formula, the problem can be further described as:(2)$$W^{*}=\underset{w}{\arg\max}\frac{p(O|W)p(W)}{P(O)}$$

Since the occurrence probability of the observed sample sequence p(O) has no effect on the solution of the optimal word sequence $W^{*}$, the problem can be simplified to:(3)$$W^{*}=\underset{w}{\arg\max}p(O|W)p(W)$$

The formula p(O|W) represents the likelihood of the speech feature vector O and the given word sequence W, which is described by acoustic model in the speech-recognition system. p(W) represents the prior probability of the word sequence W, which is described by language model in speech recognition.

### 2.3. End-to-End Model 

Since 2015, end-to-end models have become popular and have been applied to the field of speech recognition. As shown in Figure 6, the three components of the pronunciation dictionary, acoustic model, and language model of the traditional speech-recognition system are integrated into an E2E model, which directly realizes the conversion of input speech to output text and obtains the final recognition result. In general, the goal of end-to-end speech recognition is to establish the relationship between the acoustic input sequence and the sentence label sequence. Generally speaking, the sequence length of the former is significantly larger than that of the latter, so the end-to-end modeling task is essentially to establish a mapping relationship between sequences of different lengths. Typical E2E is divided into two categories: one is the speech-recognition model based on the CTC algorithm, and the other is the model based on the attention mechanism. According to a broad definition, all current end-to-end speech-recognition models can be regarded as Sequence-to-Sequence (Seq2Seq) models.

The innovation and convenience of the E2E model is that it can easily integrate many complicated steps of the previous traditional system into one model without the need to prepare various complex pronunciation dictionaries and acoustic language models. Major technological innovations have contributed to the successful implementation of some products and to a certain extent have determined the future development direction of intelligent speech. E2E reduces the reliance on professional speech knowledge and reduces the difficulty of building a voice recognition system. However, a pure data-driven E2E speech-recognition system requires a lot of data, and at least tens of thousands of hours of annotated speech are required to obtain the most advanced results. Therefore, for the low-resource speech recognition studied in this paper, the current E2E cannot bring better results than deep learning. Therefore, it will not be discussed on a large scale.

## 3. Feature Extraction

After the speech data are preprocessed and divided into frames, the corresponding feature parameters are extracted for each frame. Feature extraction is used to turn the speech signal into a feature parameter that the recognizer can process. Only when the feature parameter has the following three characteristics can good speech recognition rate be obtained: (1) distinction, which can accurately model speech acoustic units; (2) robustness—in the face of changes in speakers and channels, feature parameters will not be affected, and feature parameters can resist noise interference; (3) low dimensionality—while containing enough effective information, the feature dimension should be as low as possible to reduce the data scale and improve the system efficiency.

### 3.1. General Approach to Feature Extraction

Feature extraction is an important front-end technology in speech recognition. It can remove redundant information irrelevant to speech recognition and obtain important feature information useful for speech recognition. As a waveform sampling point, the speech time-domain signal cannot be directly used for recognition in general. The main problem with time domain signals is that it is difficult to find pronunciation patterns, and even very similar pronunciations may look very different in time domain. In fact, the human hearing organ recognizes sounds in the frequency domain rather than waveforms. The sound spectrum is obtained by short-time Fourier transform of the sound. Therefore, taking the frame as a unit, according to the auditory perception mechanism, adjust the amplitude of each component in the spectrum of the sound segment as needed and parameterize it to obtain a vector suitable for representing the characteristics of the speech signal, which is the acoustic feature.

In speech recognition, common acoustic features are Mel Frequency Cepstrum Coefficient (MFCC), Perceptual Linear Predictive (PLP) and Filter-bank (Fbank) [19,20]. The corresponding extraction process is shown in Figure 7. Among them, Fbank, MFCC, and PLP all use short-time Fourier transform with regular linear resolution. The Fbank feature needs to go through a Mel filter bank that simulates the human hearing mechanism, and the square of the magnitude of the power spectrum belonging to each filter is summed and then logarithmically obtained. MFCC features can be obtained by discrete cosine transform on the basis of Fbank. The extraction of PLP features is more complicated. A linear prediction method is adopted to realize the deconvolution processing of the speech signal, and the corresponding acoustic characteristic parameters are obtained. Fbank is an MFCC without discrete cosine transform, which preserves the correlation between feature dimensions. When training the GMM-HMM acoustic model, due to the limitation of computation, the diagonal covariance matrix is usually used, so the dimensions of the GMM probability density function are conditionally independent, so the MFCC feature is usually used. While training DNN-HMM acoustic models, especially CNNs, Fbank features are often used. The advantage of DNN is that it can process features containing multiple frames of data to represent richer speech context changes. Therefore, the input layer is often spliced with multiple frames before being sent to the network. Moreover, compared with GMM, DNN does not require very strict data distribution assumptions to estimate the posterior probability in HMM; especially, the feature elements of the same frame do not need to be independent of each other, so acoustic features can use more primitive features, such as Fbank.

Based on these underlying speech acoustic features, some enhanced processing methods are often taken to improve their performance. Among them, feature transformation is a main processing method. Acoustic feature transformation is an important part of ASR system, its purpose is to improve the recognition accuracy and reduce the computational complexity. Specifically, through a linear transformation, feature vectors are projected from an n dimensional space to a d dimensional subspace, resulting in new features with good discriminability [21]. In other words, feature transformation refers to converting one frame of acoustic features into another frame feature after some transformation. That is, the input of feature transformation is a frame of features, and the output is also a frame of features. When using feature transformation techniques, the same transformation is usually performed on a batch of acoustic features. Feature transformation is divided into unsupervised feature transformation and supervised feature transformation.

Among them, unsupervised feature transformation refers to feature transformation that does not rely on annotations. Commonly used unsupervised feature transformation techniques include Delta, Splicing, and Normalize. Delta is to calculate the different characteristics of the front and back frames within a certain window length and supplement it to the current features. Frame splicing refers to splicing several frames before and after into a frame feature within a certain window length. In the field of speech recognition, normalization is usually called cepstrum mean variance normalization (CMVN) because cepstrum coefficient is one of the most commonly used acoustic features in the field of speech recognition. Based on it, normalization of the mean and variance within a certain range is called CMVN. The purpose of normalization is to normalize the input acoustic features so that they conform to a normal distribution, that is, a vector with zero mean and a unit matrix with variance. Therefore, it is necessary to count the mean and standard deviation of several acoustic features within a certain range. Different selection of the range forms different normalization methods, namely, global normalization (Global CMVN) or speaker normalization (Speaker CMVN).

Supervised feature transformation estimates a set of transformation coefficients with the help of annotation information, enhances the representation ability of input features, and helps to improve the modeling ability of acoustic models. The most common representation of supervised feature transformation is to multiply the input by a feature transformation matrix. The estimation methods of feature transformation matrix in speech recognition are mainly divided into two categories: Linear Discriminant Analysis (LDA) and Maximum Likelihood Linear Transformation (MLLT). The purpose of LDA is to reduce the variance between similar features and increase the variance between different classes of features through transformation. The class here refers to the state of the acoustic model. The maximum likelihood linear transformation is a general term for a class of transformation techniques, including mean maximum linear natural regression (MeanMLLR), variance maximum linear regression (VarMLLR), and Feature-space Maximum Likelihood Linear Regression (FMLLR). Among them, MeanMLLR and VarMLLR are transformation methods for model parameters, while fMLLR is a technology for feature transformation.

In addition, in order to improve the distinguishability and robustness of acoustic features, reduce feature dimensions, and meet the actual needs of continuous speech-recognition systems, researchers have also proposed a variety of feature transformation methods [22,23,24,25,26,27,28,29,30,31,32]; the relevant research is shown in Table 1. LDA, Heteroscedastic Discriminant Analysis (HDA), Generalized Likelihood Ratio Discriminant Analysis (GLRDA), etc., can improve feature discrimination and reduce feature dimensions; Maximum Likelihood Linear Regression (MLLR), fMLLR, and Vocal Tract Length Normalization (VTLN) can eliminate speech information that has nothing to do with the recognition result, such as people or soundtrack, improving the robustness of features.

As a commonly used statistical pattern classification technology, LDA improves the resolution of feature vectors through linear transformation and compresses the information content related to classification [22,23,24,25]. Among them, Ney et al. studied the interaction between LDA and continuous Laplacian hybrid density modeling method in the recognition task of 12,000 German words, with little overlap between training and test vocabulary. Experiments show that using the LDA acoustic feature transformation method reduces the recognition error rate by one-fifth compared to the case without using the LDA acoustic feature transformation method [22]. LDA is suitable for classifier models with equal variances in the class distribution. As a model-based generalization of LDA, HDA is more suitable for the case where the class distribution is heteroscedastic [33,34,35]. Kumar et al. removed the equal variance assumption in the parameter model and promoted LDA under the maximum likelihood framework to deal with heteroscedasticity. Specifically, the EM algorithm and dimensionality reduction transformation are used to estimate the parameters of the heteroscedastic Gaussian model, where each Gaussian distribution is modeled by a full covariance matrix. When the context size is 4, on the TI-DIGITS database, compared with the baseline system using LDA, the error recognition rate of using HDA is relatively reduced, by 12% (67%→59%) [26]. In [27], based on the likelihood ratio test, Lee et al. proposed a new discriminant feature transformation—GLRDA. Under the assumption that the class distribution is not homoscedastic, it tries to find a low-dimensional feature subspace by making the most confusing situation described by the null hypothesis not occur as much as possible. In [28], Leggetter et al. proposed MLLR. This method uses a set of regression-based transformations to adjust the mean parameters of the Hidden Markov Model (HMM) and uses the adjusted parameters for the new speaker. Each transformation in the experiment is applied to some HMM mean parameters and estimated from the corresponding data. Through this conversion and data sharing, a small amount of adaptive data are obtained, and the system performance is improved. Subsequently, Leggetter et al. experimented with this speaker-adaptation technology on the ARPA RM1 database. The results show that even if the data are as short as 11s, the adaptive performance has a good experimental effect, and as more data are used, the adaptive performance will improve. For example, compared to speaker-independent systems, using 40 adaptive utterances in supervised adaptive and unsupervised modes can reduce WER by 37% and 32% [28]. 

In order to reduce the mismatch between the model and the acoustic data from a specific speaker, Woodland et al. proposed FMLLR, also known as constrained MLLR [29,30,36], that is, the Gaussian mean and variance share a linear transformation matrix. This method not only works well after obtaining a few sentences of adaptation data but also works reliably in various adaptation modes. Even this method does not require any prior information about the speaker (or environment) type distribution. However, one disadvantage of this technique is that since there is no model for speaker changes, any acoustic mismatch will be modeled. Based on this shortcoming, a number of methods have been proposed to prevent overtraining. As a commonly used method to reduce inter-speaker variability, VTLN has received extensive attention from researchers. The principle of VTLN is to perform appropriate frequency distortion on the frequency spectrum of the speech frame to reduce the frequency spectrum changes between different speakers [31,32]. However, because all acoustic categories do not show similar spectral changes, different acoustic categories need to use different frequency warping factors in the research. In other words, the use of single-frequency distortion in the entire pronunciation will have a certain impact on the recognition results of the experiment, leading to an increase in WER.

### 3.2. DNN-Based Approach to Extract Deep Acoustic Features

Common acoustic features include MFCC and PLP. However, in a low-resource environment, these traditional shallow features are not robust enough to meet the requirements of system modeling. In order to obtain more robust feature parameters, traditional shallow features are usually non-linearly transformed to extract deep features. The predecessor of deep learning is Artificial Neural Network and Multi-Layor Perceptron (MLP). In low-resource speech recognition technology, currently widely used is the use of MLP or Deep Neural Network (DNN) to extract deep features. DNN is also an MLP in essence. However, due to the limitation of theoretical research level and hardware computing power in the past, the MLP used by researchers usually has a small number of hidden layers (1~2), and the network parameters are randomly initialized. Therefore, MLP is now generally used to refer to such a Neural Networks. The DNN is expanded on the original MLP structure, its hidden layers generally number more than 5; the category target granularity of the output layer is smaller than the original, and the number of categories is greater. Moreover, it usually uses some effective pre-training methods to replace the initialization of the original random network parameters, effectively guaranteeing the speed and accuracy of DNN training. In addition, DNN has different structures, including CNN, LSTM, and TDNN. 

At present, the commonly used deep features include Tandem feature and Bottleneck (BN) feature. They are extracted from the output layer and hidden layer of MLP or DNN, respectively. Both Tandem feature and BN feature are neural network feature parameters. The similarity is that, compared with the traditional feature parameters, they have higher distinguishability and robustness and present a certain degree of language independence. The difference between them is that the Tandem feature uses the computing power of the neural network to enhance the discrimination of the original feature, while the BN feature is a powerful non-linear dimensionality reduction. Numerous studies have shown that in low-resource speech recognition, you can borrow labeled data in multiple languages to assist in training the weights of neural networks, thereby improving the performance of the speech-recognition system. In addition to borrowing multilingual data to assist in training the neural network, different posterior features or acoustic features can also be spliced and fused to enhance the recognition performance of features. Table 2 summarizes the research on extracting deep features in recent years.

Specifically, in [37], Tandem features are obtained from the logarithmic posterior probability of the MLP output layer using Principal Component Analysis for de-correlation and dimensionality reduction and then spliced together with the original shallow features. Experiments show that the Tandem feature is significantly better than the baseline system. When based on the same cepstrum feature, an average relative WER reduction of 35% can be achieved under multiple test conditions. BN features are extracted through a neural network with a special structure. The network has a hidden layer with a relatively small number of nodes, and the output of the hidden layer is the BN feature. In [38], BN features are derived from the hidden layer of MLP. The experiment has 5 layers of MLP, and the middle layer has a bottleneck. The five-layer MLP is used because the structure has sufficient ability to extract internal features and can efficiently classify them. After training the neural network, the output of the bottleneck layer is used as the feature of the Gaussian Mixture Model Hidden Markov Model (GMM-HMM) recognition system. Experiments conducted on the meeting recognition task defined in the NIST RT ‘05 evaluation show that when the feature size is 35, the relative WER of the BN feature recognition system extracted from the five-layer MLP decreased by 0.7% (25.6%→24.9%) compared to the baseline system, where the probability feature is the input acoustic feature. 

Initially, BN features were regarded as dependent on language, until the speech team of Brno University of Technology proposed a language-independent BN feature extraction framework [39]. The framework is a five-layer MLP with a sigmoid hidden unit, a linear bottleneck, and several output layers. Each language has its own weight and softmax function. Because each output layer is modeled by each language separately, while the hidden layer is jointly modeled by all languages and trained for all languages, BN features is not biased to any language, that is, the final BN features are considered to be language-independent. Based on this framework, the researchers used the eight languages of Czech, English, German, Portuguese, Spanish, Russian, Turkish, and Vietnamese in the GlobalPhone data set to do two sets of comparative experiments. (1) The monolingual bottleneck MLP is trained separately for each language as the baseline. (2) The proposed language-independent BN feature extraction framework is used with eight output layers for comparison. Experimental results show that for different languages, WER has different degrees of reduction. Among them, the WER of Czech was reduced from 19.7% to 19.3%; WER of English was reduced from 15.9% to 14.7%; WER of German was reduced from 25.5% to 24.0%; WER of Portuguese was reduced from 27.2% to 25.2%; WER of Spanish was reduced from 23.2% to 22.6%; Russian WER decreased from 32.5% to 31.5%; Turkish WER decreased from 30.4% to 29.4%; and Vietnamese WER decreased from 23.4% to 24.3% [39]. Moreover, the experimental results show that the BN features extracted using this language-independent BN feature extraction framework are always better than single-language BN features.

Yebo Bao et al. used the distinguishing characteristics of BN features extracted by DNN and used DNN as a BN feature extractor to achieve the purpose of decorrelating the long feature vectors connected by several consecutive speech frames. They combined with traditional GMM-HMM to form the Tandem system. The experimental results of Yebo Bao et al. in the 70-h Mandarin transcription task and the 309-h Switchboard task showed that the traditional GMM-HMM using BN features can produce performance comparable to the Deep Neural Network Hidden Markov Model (DNN-HMM) [40].

In addition to the above-mentioned commonly used deep features Tandem and BN, in low-resource speech recognition, labeling data in multiple languages can also be borrowed to assist in training the weights of the neural network, thereby improving the performance of the speech-recognition system. The specific implementation method is to first train the MLP with high-resource data and then pass the target speech feature forward to obtain a feature suitable for low-resource speech recognition. Since this method reduces the requirement on the quantity of task data, it is suitable for low-resource situation [41,42]. Stolcke et al. used MLP trained by English speech data for feature extraction in the continuous speech recognition of Mandarin Chinese and Mediterranean Arabic, and the recognition rate was improved relative to the baseline system, verifying the effect of the method of borrowing the labeling data of multiple languages to assist the weight training of neural network in continuous speech recognition [43]. In DNN, there is not only a nonlinear layer but also a linear layer. The linear layer is the softmax layer, and the nonlinear layer is in the hidden layer. Before Microsoft Research proposed the Shared-Hidden-Layer Multilingual Deep Neural Network (SHL-MDNN), most of the feature conversions determined by the hidden layer in the system were learned from monolingual data. The SHL-MDNN framework proposed by Microsoft Research is shown in Figure 8. In SHL-MDNN, the input layer and hidden layer are shared by all languages, the softmax layer is unique to a specific language, and the shared layer is called the language general feature extractor. That is, the SHL extracted from SHL-MDNN is a good feature extractor for deep feature extraction. Microsoft Research evaluated the SHL-MDNN model on Microsoft’s internal speech recognition task, using DNN individually trained in four European languages, including French (FRA), German (DEU), Spanish (ESP), and Italian (ITA) as the baseline system. The experimental results showed that the WER of the proposed SHL-MDNN decreased uniformly, and the WER of French (FRA) decreased by 1% (28.1%→27.1%); the WER of German (DEU) decreased by 1.3% (24.0%→22.7%); the WER of Spanish (ESP) decreased by 1.2% (30.6%→29.4%); and the WER of Italian (ITA) decreased by 0.8% (24.3%→23.5%). It can also be seen that the proposed SHL-MDNN model is superior to monolingual DNN [44].

It can be known from Table 2 that in addition to borrowing multilingual data to assist in training the neural network, different posterior features or acoustic features can also be spliced and fused to enhance the recognition performance of features. Karafiat et al. spliced the Mel filter group, three different pitch features, and the fundamental frequency variance and other shallow features; they then used them as the input features of the stacked bottleneck neural network to extract the deep features. The stacked bottleneck neural network consists of two neural networks: the output of the first bottleneck layer is adjusted so that the stack exceeds 21 frames, and then it is down-sampled and used as the input of the second neural network. The second neural network also has a bottleneck layer, and the output of the bottleneck layer is the deep feature. Experiments on BABEL 2014 Tamil limited data package show that the WER of the baseline DNN system based on PLP features as input is 74.9%, the WER of the deep feature system based on feature splicing is 73.8%, and the deep feature system based on splicing shallow features reduces the WER of the system by 1.1% [45]. In the literature [46], Thomas et al. used long and short-term complementary features to train MLP separately, extract the BN features corresponding to each feature, and then splice them to obtain a recognition feature that contains more speech information. Experiments show that this splicing feature reduces WER by 30% in low-resource speech recognition tasks. The research team of Johns Hopkins University in the United States first used high-resource languages to train MLP and extract the posterior features of high-resource languages; they then spliced the features as complementary information with low-resource acoustic features and sent the feature vector into a new MLP to train. Finally, the posterior features of the target language are obtained and taken as the final acoustic features. Compared with the baseline system that directly uses the posterior features of the target language as the final acoustic features, the team’s WER from the splicing features trained with one hour of English telephone conversational speech (CTS) decreased by about 11% [47].

Due to the rapid development of deep learning technology in recent years, in addition to the above-mentioned deep feature extraction methods, low-rank matrix factorization (LRMF), multi-language-training and parameter-sharing convolutional neural network (CNN), extraction of amplitude modulation (AM) features, and learning of language-invariant bottleneck features from adversarial end-to-end models have all entered people’s field of vision. Specifically, the MIT Computer Science and Artificial Intelligence Laboratory proposed to use LRMF to extract the BN features of the DNN decomposition weight matrix. The experiment proved that the Tandem system built by this method and the DNN-HMM system have similar recognition rates [48]. In [49], Miao et al. proposed to use CNN to extract convolutional network neurons as acoustic features (the dimension of convolutional network is thousands of dimensions), which outperformed the features extracted by DNN of the same dimension. Sailor et al. verified that the AM features extracted from the gamma tone filter bank perform well on the low-resource Indian language Gujarati. The team used the Recurrent Neural Network Language Model and the Time-Delayed Neural Network (TDNN) acoustic model to make WER reduce the test set and blind test set by 1.89% and 2.24%, compared with the FBANK baseline [50]. Yi et al. proposed to use adversarial multi-language training to extract the general bottleneck features of low-resource languages. The results show that the proposed method is effective, but there are still some limitations [53,54]. Its limitations are shown by the fact that (1) the language confrontation model is trained with the cross-entropy loss function, but it is not clear whether the model trained with the connectionist temporal classification (CTC) loss function is effective; (2) the input features of a few frames do not contain too much language information; and (3) shared and private features may be similar. In order to solve the above problems, Jiangyan Yi et al. were inspired by the end-to-end acoustic model of CTC [55,56] and proposed to learn from the adversarial end-to-end model to learn language invariant bottleneck features. They used the BLSTM model with CTC loss function on the IARPA Babel data set to train the adversarial bottleneck model. Experimental results show that the target model trained with the proposed language invariant bottleneck feature is better than the target model trained with traditional multilingual bottleneck features, and the relative WER is reduced by up to 9.7% [51]. 

Normally, the unit used for modeling is phoneme. However, compared with the phonemes of each language, the articulatory feature (AF) is more representative of the articulatory organ that produces speech. AF is a characteristic representation of the basic properties of speech signals in speech production. It depends more on the voice of the speaker, so it is a language-independent modeling form [57,58,59,60,61]. AF is robust to changes in channels and speakers and has cross-language portability [62,63,64,65]. Therefore, AF simulates the contextual information in speech better than traditional features and performs very well in unfavorable acoustic environments [62]. The Stacked Bottleneck Feature (SBF) framework uses a cascade of two networks with a bottleneck layer, which provides two abstraction levels for feature extraction [66]. The Indian Institute of Technology proposes a feature extractor model that extracts SBF and AF to form a connected feature vector, which will be used as the input feature vector of the acoustic model. The feature extraction model is shown in Figure 9a, and the pronunciation classifier used to extract AF is shown in Figure 9b. Because a large number of data are required to build an efficient pronunciation classifier, this experiment is used to train Bidirectional Long Short-Term Memory (BLSTM) pronunciation classifier by pooling data from available low-resource Indian languages (Gujarati, Tamil, and Telugu). The feature extractor used to extract SBF is shown in Figure 9c. The feature extractor has two DNNs. The first network is a multilingual DNN, which is trained by pooling data in all three languages. Each language has its own softmax layer on the output. The network acts as a multilingual feature extractor. Another DNN is built on the first multilingual feature extractor. In order to bias the second network towards the target language, the team extracted the bottleneck features of the target language from the bottleneck layer of the first network and used these features to train the second network. Therefore, the model has an initial network that captures language-independent features and a second network that adapts to the target language. The Indian Institute of Technology uses low-resource Indian languages (Gujarati, Tamil, and Telugu) for comparative experiments. The baseline system is a TDNN trained with traditional MFCC features. The experimental results show that the WER of Gujarati and Telugu consistently decreases, but the WER of Tamil does not decrease. Among them, the WER of Gujarati language dropped from 14.61% to 14.11%, and the WER of Telugu language dropped from 21.44% to 19.80%. The team concluded from the experimental results that the proposed combination of AF and SBF is an improvement over traditional features [52]. 

## 4. Acoustic Model

Acoustic model is one of the most critical steps in acoustic processing in speech recognition. It describes the mapping relationship between speech acoustic units and their feature vector sequences. Since this section studies acoustic modeling in low-resource environments, the most important point is to address the lack of training data. At present, there are two basic ideas in acoustic modeling: one is to implement auxiliary training by introducing data from other languages to the target acoustic model, that is, to borrow data from a resource-rich language to improve the training effect of the target acoustic model and ultimately improve the recognition rate of low-resource speech-recognition systems. The other is to compress the parameters of the acoustic model, that is, to reduce the model parameters.

### 4.1. General Approach to Acoustic Modeling

Through the acoustic model, the speech recognition acoustic unit can be estimated according to the feature vector sequence to complete the recognition and conversion of the speech signal. Since the speech signal is a non-stationary signal with short-term stationarity, according to this characteristic, the HMM method is commonly used to acoustically model the speech signal. The objects to be modeled are generally smaller units than words, such as phonemes. HMM can estimate pronunciation probability distribution very well. A typical three-state HMM is shown in Figure 10.

HMM contains two stochastic processes: one process produces the state sequence $Q=\left.\left\{q_{1},\ldots q_{t},\ldots q_{T}\right.\right\}$ (plus the initial state $q_{I}$ and $q_{E}$) and the other process produces the observation sequence $X=\left.\left\{x_{1},\ldots x_{t},\ldots x_{T}\right.\right\}$, in which the state sequence Q is not directly observable.

HMM needs to solve three problems. First, the forward algorithm is used to calculate the likelihood probability that HMM generates a series of observation sequences X to complete the model evaluation; secondly, the Viterbi decoding algorithm is used to calculate the most likely hidden state sequence given the HMM observation sequence; and finally, GMM is introduced for model training. In GMM-HMM, it is assumed that the emission probability of HMM is obtained by superposing a series of Gaussian distributions.

### 4.2. DNN-Based Approach to Build Acoustic Model

With the development of machine learning technology and hardware computing capabilities, researchers have proposed a variety of better-performing classification models, which have been successfully applied to the acoustic modeling process. Compared with traditional models, deep learning-based acoustic models are more discriminatory to target data. In multi-language training, with the increase of training corpus data, the number of parameters basically remains at the same order of magnitude, and the small increase of parameters mainly comes from the increase of nodes in the classification output layer brought by multi-language targets. As shown in Table 3, relevant researchers have done the following research on acoustic modeling in low-resource environments.

When the data of the target language are scarce, the multilingual training data can be used to improve the recognition rate of the target language. The Idiap Institute of Martini in Switzerland and the Lausanne Institute of Technology proposed the Kullback–Leibler Hidden Markov Models (KL-HMM), based on Kullback–Leibler Divergence (KLD), and performed the experiments on the Greek Speech HDAT (II) dataset [69]. In current research, the latest low-resource speech recognition technology mostly uses phoneme modeling. However, because of the high pronunciation variability of words (within the same language) and the variability of the acoustic realization of the same phoneme (within and between languages), it is still a challenging task to train phoneme acoustic models. In response to the problems of phoneme modeling, the Idiap Institute of Martini in Switzerland and the Lausanne Institute of Technology proposed a random phoneme space conversion technology that allows for the posterior probability of the conditional source phoneme (with acoustics as a condition) to be converted into posterior probability of the target phoneme. The proposal of this technology means that the source phoneme and target phoneme can be in any language and phoneme format (such as the International Phonetic Alphabet) [70]. In addition, because of multi-language training, the posterior probabilities of phonemes of different language modeling units do not match. To solve this problem, Thomas et al. of the Johns Hopkins University team formed a Confusion Matrix by forward propagation of low-resource target features, and the Maximum Mutual Information criterion was used to establish a one-to-one mapping between the low-resource modeling units and the high-resource modeling units [71].

Compared with the DNN model, the deep CNN model has better robustness and generalization ability [72,88]. In the past, Markus et al. have confirmed that multilingual training is one of the important techniques used in feature extraction when the amount of available training data is limited [89,90,91,92]. Researchers have described a sequence to sequence neural network that generates speech waveforms directly from text input. This model can be directly optimized with maximum likelihood, without using intermediate, manually designed features or additional loss terms. Inspired by this idea, researchers can also construct a sequence to sequence neural network that directly generates text output from speech waveforms [93]. Martin et al. extended this idea to the BLSTM acoustic model and demonstrated the significant effect of multilingual training on low-resource languages. In this experiment, all multi-language models are trained using the block-softmax output layer, which is composed of softmax for each language [73]. The process of porting the multilingual model to the target language by Martin et al. can be described as the following steps: (1) the final multi-language layer is stripped and replaced with a target language-specific layer for random initialization; (2) the new layer is trained for 8 epochs with the standard learning rate, while the rest of the NN is fixed; (3) fine-tune the entire NN for 10 epochs and set the initial value of the learning rate plan to 0.5 of the original value. The experiment used the languages in the Babel project as a control experiment and selected three languages—Javanese, Amharic, and Pashto—as the target languages. Compared with the BLSTM baseline system that only uses monolingual training, the WER of the BLSTM that uses the 24 languages in the Babel project is reduced to varying degrees. Among them, the WER of Javanese dropped from 54.0% to 49.2%, the WER of Amharic dropped from 44.0% to 39.6%, and the WER of Pashto dropped from 48.7% to 46.0% [73]. The literature [74] proposed Shared-Hidden-Layer Multilingual Long Short-Term Memory (SHL-MLSTM). Compared with the SHL-MDNN trained in six languages, the WER of SHL-MLSTM can be relatively reduced by 2.1–6.8%.

India is a diverse and multilingual country. The languages spoken by inhabitants of over a billion people vary widely. However, the lack of resources in transcribing speech data, speech pronunciation dictionaries, and text collection has hindered the development and improvement of Indian ASR systems. Fortunately, the Interspeech 2018 Special Session—Low-Resource Speech Recognition Challenges in Indian Languages—has started working on low-resource speech recognition in Indian languages. We can be sure that because of the existence of this conference, more people will be attracted to solving low-resource speech recognition challenges in Indian languages. Fathima et al. presented a system with the second highest performance at this conference. The system submitted by Fathima et al. mainly consists of two techniques: (1) when training an acoustic model, multilingual training has advantages over single-language training; (2) when training with multiple languages, language-specific information is used in conjunction with the acoustic model for decoding, which can achieve better decoding than mixing all language models. The languages involved in the system include Tamil, Telugu, and Gujarati, of which Tamil and Telugu belong to the same Dravidian language family, while Gujarati belongs to Indo-YaLi’an language. Each language has 40 h of training data and 5 h of test data. Fathima et al. built a multilingual TDNN system that uses combined acoustic modeling and language-specific information to decode input test sequences. The overall architecture of the constructed system is shown in Figure 11. The source languages here are Gujarati, Tamil, and Telugu. The training phase uses approximately 120 h of combined audio data along with corresponding transcripts. Using MFCC without cepstral truncation, a speaker-dependent GMM-HMM system is constructed using FMLLR features. The alignment obtained from this model is used for LF-MMI training of the chain TDNN. A combined dictionary covering words present in the training set of all three languages is used. During the testing phase, features are extracted from the test audio and decoded using language-specific language models to obtain output transcriptions for each language. Fathima et al. conducted three sets of experiments: (1) a TDNN baseline system trained with a single language; (2) this is trained with three languages and decoded with a three-language mixed language model or with three language-specific language models; (3) it is then train with two languages belonging to the same Dravidian language family and decoded with a mixed language model of the two languages or with a language-specific language model of the two languages. For Gujarati: the WER of TDNN trained in a single language was 12.7%. The WER of the language model decoded by three languages was 11.95%. The WER of language model decoding with three languages’ training and using three specific languages was 11.58%. For Tamil: the WER of TDNN trained in a single language was 16.35%. The WER was 17.32% when the language model was trained in three languages and mixed with three languages. The WER of language model decoding with three languages and three specific languages was 16.79%. The decoding WER of Tamil and Telugu, which belong to the same Dravidian language family, was 16.46%. Trained in Tamil and Telugu, which belong to the same Dravidian family, and using the language models of these two specific languages, the WER of decoding is 16.47. For Telugu, the WER of TDNN trained in a single language was 18.61%. The WER of language model decoding was 18.84% when trained in three languages and used a mixture of three languages. The WER of language model decoding with three languages and three specific languages was 17.94%. The decoding power of Tamil and Telugu, which belong to the same Dravidian language family, was 17.69%. Trained in Tamil and Telugu, which belong to the same Dravidian family, and using the language models of these two specific languages, the WER decoding rate is 16.95% [75]. Likewise, the BUT team created a speech-recognition system for the 2018 Indian Language Low-resource Speech Recognition Challenge. The acoustic modeling of the system submitted by the BUT team is mainly aimed at TDNN. The system submitted by the team mainly consists of two techniques: (1) a model with a low-rank TDNN architecture trained using the LF-MMI target and (2) a transfer learning approach to adapt a multilingual TDNN model. The team conducted three sets of experiments: (1) a monolingually trained TDNN as the baseline system; (2) a monolingually trained low-rank DTNN. The main difference between this system and the baseline system is the addition of a bottleneck linear transformation layer and skip connections after each affine transformation of the batchnorm ReLU layer. The team assumes this is a low-rank decomposition of TDNN on each ReLU + linear layer pair. (3) In transfer learning, the source model is trained on a large corpus, then the weights of the hidden layers are transferred to a smaller target dataset and retrained for similar or different tasks. In this approach, both models must be trained with the same objective function. In the BUT team’s experiments, the source and target models are multilingual and monolingual low-rank TDNNs trained with LF-MMI targets. The BUT team transferred all layer models from the pretrained source and retrained the last layer for two epochs with a higher learning rate using the target labels. The learning rate of the transfer layers (excluding the last layer) is reduced by a factor of 0.25 of the initial learning rate. New alignments and grids are generated using a multilingual low-rank TDNN model. In transfer learning, the last layer is not transferred because source and target use different phoneme sets and trees, but in their experiments, all layers are transferred because they use the same context-dependent tree of the source network. Experimental results show that the model trained with the low-rank TDNN architecture using the LF-MMI objective outperforms the traditional TDNN. Using a transfer learning method to fit a multilingual TDNN model can further improve the recognition rate. The system submitted by the BUT team achieved WER of 13.92% (Tamil), 14.71% (Telugu), and 14.06% (Gujarati) [76].

The Indian Institute of Technology uses data and model parameters from other high-resource languages to improve the low-resource language acoustic model. The specific implementation method is to use the method of transfer learning to let the complex high-resource model guide the training of the low-resource model. In transfer learning, the output layer of DNN has undergone multilingual DNN training and a separate softmax layer training. We call this method blocksoftmax [94]. The blocksoftmax framework is shown in Figure 12a. The training data of high-resource and low-resource languages simultaneously perform a separate softmax activation for each language in the output layer, and the cross-entropy error of each language data are only generated by the corresponding softmax layer. In the experiment, an additional KLD-based constraint technology is used in the blocksoftmax framework to overcome the data sparse problem of DNN-based acoustic model modeling. The proposed model framework is shown in Figure 12b. The KLD-based constraint avoids the problem of overfitting by acting as a regularization term in the loss function during blocksoftmax multilingual training using low-resource and high-resource languages. The additional KLD-based constraint is another output target from the high-resource model probability distribution in terms of low-resource data. These goals are obtained by forwarding low-resource data through the high-resource DNN model. Being different from traditional hard targets, each target has multiple non-zero values, and these targets are called soft labels. Compared with the low-resource model, the deviation of the posterior distribution of the high-resource model is closer to its true distribution, so this additional regularization term forces the output error of the low-resource model to be basically consistent with its true distribution. The research team used three low-resource Indian languages (Hindi, Tamil, and Kannada) to conduct experiments. Each language has a 50-h high-resource data set and a 10-h low-resource data set. The team conducted three comparative experiments: (1) a DNN baseline system using 10 h of monolingual language, (2) Blocksoftmax system using 50-h high-resource data sets in both languages and 10-h low-resource data sets in test languages, and (3) Blocksoftmax system with additional KLD using 50-h high resource data sets in both languages and 10-h low-resource data sets in test languages. Experiments show that the Blocksoftmax system with additional KLD proposed by the Indian Institute of Technology has the best performance, followed by the Blocksoftmax system, and the monolingual DNN baseline performs the worst. Specifically, compared with monolingual DNN baseline system and Blocksoftmax system, the WER of Hindi decreased by 4.09% (16.14%→12.05%) and 0.45% (12.50%→12.05%), respectively; the WER of Tamil decreased by 3.36% (15.76%→12.40%) and 0.09 (12.49%→12.40%) respectively; and the WER of Kannada decreased by 1.06% (7.41%→6.35%) and 0.12% (6.47%→6.35%), respectively [77].

In 2020, the National Institute of Standards and Technology (NIST), in collaboration with the Intelligence Advanced Research Projects Activity (IARPA), publicly challenged ASR technology for low-resource languages over challenging data types (conversational telephone speech). OpenASR20 Challenge targets 10 under-resourced languages: Amharic, Cantonese, Guarani, Javanese, Kurmanji Kurdish, Mongolian, Pashto, Somali, Tamil, and Vietnamese. A total of 9 teams from 5 countries participated in the competition, with 128 valid submissions. The OpenASR20 Challenge offers two different training conditions: Constrained (CONSTR) and Unconstrained (UNCONSTR). For any language processed, participants must submit constrained training conditions, although unconstrained training conditions are optional. As the name suggests, constrained training conditions limit training data resources to allow for better cross-team comparisons. In this case, the only speech data allowed for training is the 10-h training set specified by NIST in the language being processed. Additional textual data in any language, whether from a provided training set or publicly available resources, allows for training under restricted training conditions. Under unconstrained training conditions, participants can use speech data beyond the provided 10-h training set, as well as other publicly available speech and text training data from any language. This case allows performance gains to be measured from additional training data. Any such additional training data must be specified in the system description. Participants may not employ native speakers for data collection, system development, or analysis of any training conditions. With tightly constrained training data, results show a high (WER) overall, with best results ranging from 0.4 to 0.65, depending on the language [95]. The TNT team participated in the OpenASR20 Challenge under the restricted conditions of Cantonese and Mongolian. Since the NN-HMM hybrid acoustic model proved to be more promising in terms of ASR performance than the E2E structure under certain under-resource conditions, the hybrid acoustic model was adopted throughout the restricted conditions [96]. There are 3 main technologies included in the system submitted by the TNT team: (1) the CNN-TDNN-FA architecture is proposed as the main acoustic model, which introduces a self-attention network (SAN) in the combination of CNN and TDNN-F, in order to learn more location information from the input; (2) adopt a self-training strategy in the front end of the hybrid system to obtain a more efficient representation with only 10 h of speech data; and (3) combine various data augmentations to obtain additive improvement. The CNN-TDNN-FA network contains a total of 11 TDNN-F blocks (9 before the SAN, 2 after the SAN), with a hidden dimension of 768 and a bottleneck dimension of 160. They start with six convolutional blocks and concatenate i-vectors and hires-MFCCs as input. Because it is aimed at low-resource situations, the team combines various data augmentations to obtain additive improvements, such as velocity perturbation, Spec-Augment, Wav-Augment, added noise, and reverberation [97,98]. In the end, the TNT team obtained a WER of 0.483 on the Cantonese development set and a WER of 0.402 on the evaluation set. Likewise, a WER of 0.524 was obtained on the Mongolian dev set and 0.449 on the evaluation set [78]. The Tallinn University of Technology (TalTech) team participated in OpenASR20’s restricted training conditions in all 10 languages. The scores they submitted were the best of any team in six out of ten languages. The system submitted by the TalTech team consists of two main techniques: (1) for each language, the researchers trained two acoustic models: one on clean velocity perturbed data and the other on noise augmented data. The structures of the models are all CNN-TDNNF and are trained according to the Kaldi “chain” model training method; (2) since the evaluation data are not segmented into utterances, the TalTech team trained a voice activity detection model to detect speech regions in the test data. SpecAugment was used during the training of the first model, which was trained for 20 epochs. For noise augmentation, the standard multi-condition training method implemented in Kaldi was used: four replicas were made from clean velocity perturbation data. Because there are now four times as much training data, the second model is only trained for five epochs. Since the evaluation data were not segmented into utterances, the TalTech team trained a voice activity detection model to detect speech regions in the test data. Figure 13 gives a visual representation of the decoding pipeline. The decoding process has the following steps: first pass→LM adaptation→RNNLM (fwd)→RNNLM (bwd)→combination. For the Gurani development set, the WER after each decoding step for the speed perturbed data is 44.0%→43.6%→42.1%→41.8%→40.3%; the WER for the noise-enhanced data after each decoding step is 44.1%→43.9%→42.3%→41.9%→40.3%. For the Javanese development set, the WER after each decoding step is 56.2%→55.7%→54.9%→55.2%→53.7%, while the WER for the noise-enhanced data after each decoding step is 56.1%→55.7%→54.9%→55.2%→53.7%. For most languages, the trends are similar for the two languages: although the absolute WERs across languages are very different, first-pass decoding with either of the two acoustic models yields similar WERs. In terms of first pass results, lattice re-scoring and combining lead to 5–10% relative improvement [79].

NSYSU-MITLab participated in the Formosa Speech Recognition Challenge 2020 (FSR-2020), which focused on the low-resource language Taiwanese (Taiwanese Hokkien) [99]. In the experiments, the acoustic model used is DNN-HMM. In the DNN-HMM system, the team implemented a time-constrained self-attention mechanism and decomposed time-delay neural networks for DNN front-end acoustic representation learning. The system submitted by the NSYSU-MITLab team mainly includes three technologies: (1) the proposed TRSA-Transformer acoustic model architecture, (2) the proposed TRSA Transformer + TDNN-F acoustic model architecture, and (3) the proposed TRSA-Transformer + TDNN-F + Macaron FNN acoustic model architecture. Figure 14 shows three different acoustic model architectures. The acoustic model starts with a TDNN layer with 816 output channels, a kernel size of 3, and a dilation rate of 3. A six-layer Transformer-linked architecture called TRSA-Transformer is adopted after the TDNN layer. Similar to Transformer, TRSA-Transformer consists of a multi-head attention layer and FFN, but the self-attention mechanism is replaced by time-restricted self-attention (TRSA). TRSA focuses on local context rather than global context. After the TRSA layer is FFN, which has an internal dimension of 1024. Batch normalization and residual connections are used after FFN. The last layer of the acoustic model is a 256 linear bottleneck with semi-orthogonal constraints, which helps in many cases. The structure of TRSA-Transformer is shown on the left side of Figure 14. TDNN-F can effectively reduce the number of parameters while maintaining the modeling ability and training stability. The team inserted a TDNN-F layer after the TRSA layer to further improve the performance of the acoustic model. The architecture of TRSA Transformer + TDNN-F is shown in the middle of Figure 14. Inspired by [100,101], researchers implemented a Macaron-like structure using two half-step FFNs: one before the TRSA layer and the other after the TDNN-F layer. This architecture is more efficient than a single FNN with the same number of parameters. The architecture of TRSA-Transformer + TDNN-F + Macaron FNN is shown on the right side of Figure 14. In FSR-2020, we obtained the best word error rate of Taiwanese Hokkien recommended characters of 43.4%, and the best syllable error rate of Taiwanese Hokkien Lomazi Pinyin was 25.4% [80].

Aiming at the problem of poor recognition rate of low-resource speech recognition due to the different types of training data and test data, Srikanth et al. developed a multi-type speech-recognition system for low-resource languages, in which the training data are mainly conversational speech. However, test data can be one of the following types: newscasts, feature broadcasts, and conversational speeches. In general, speech recognition for low-resource languages is usually developed by adapting a pretrained model to the target language. When the training data are mainly from one type and limited, the performance of the system on other types suffers. To deal with this out-of-domain scenario, Srikanth et al. propose multi-task adaptation (MTA) using auxiliary conversational speech data from other languages as well as target language data. To contrast with MTA, we refer to the traditional adaptation of the pretrained model to the target language as single-task adaptation (STA). In this study, Srikanth et al. used the multilingual LF-MMI-trained CNN-TDNN-F architecture (Convolutional Neural Network and Decomposed TDNN) obtained from the Babel and MATE RIAL datasets for 18 languages as a pre-trained model [102]. Experimental results show that a relative improvement in WER of up to 7.1% can be achieved by replacing STA with MTA on the MATERIAL dataset [81]. The application of deep neural networks to acoustic modeling tasks for automatic speech recognition resulted in significantly lower ASR word error rates, enabling the use of the technology to interact with smartphones and personal home assistants in high-resource languages. However, developing an ASR model of this caliber requires hundreds or thousands of hours of transcribed speech recordings, which presents challenges for the vast majority of the world’s languages. In order to save huge time and money costs, Ethan et al. wanted to verify whether there is a model that works for all low-resource languages. If this assumption holds true, the identification of low-resource languages will become easier in the future. In this experiment, Ethan et al. investigate the utility of three different architectures that have previously been used to train speech recognition in languages with limited resources. The three structures are as follows: Subspace Gaussian Mixture Model (SGMM) with Maximum Mutual Information (MMI), DNN with State-level Minimum Bayesian Risk (sMBR) criterion, and WireNet. WireNet is a novel fully convolutional ASR architecture developed for Seneca, a resource-rich and endangered language native to the United States and Canada. The main architectural feature of WireNet is a stack of Inception and ResNet-style blocks with wide filter widths to model the temporal properties of audio [103]. Ethan et al. trained and tested these systems on five publicly available ASR datasets of varying types and spellings, generated under various conditions using different speech collection strategies, practices, and devices. Despite the comparable sizes of these corpora, we found that none of the ASR architectures outperformed all others. Additionally, word error rates varied widely. The results of the experiments show that it is important to consider language-specific and corpus-specific factors and try multiple approaches when developing speech-recognition systems for languages with limited resources. No single model works for all low-resource languages [82].

### 4.3. GMM-Based Approach to Building an Acoustic Model

Regarding the shortcomings of insufficient training data relative to parameters in traditional GMM modeling methods, researchers have conducted a significant amount of research to solve the above problems. (1) Directly regularizing the traditional GMM can reduce the parameter amount of the model, thereby obtaining a more compact acoustic model. (2) SGMM acoustic modeling method of compressed model parameters.

In machine learning, in order to deal with low-resource model training, regularization terms are usually added to the objective function to punish complex models or impose prior knowledge. Sivaram et al. achieved the purpose of analyzing the internal hidden layer in MLP by adding a sparse regularization term to the cross-entropy cost function and minimizing the joint cost function to update the network parameters [104]. Because Gaussian likelihood is evaluated as a quadratic function determined by the inverse covariance matrix, Gopinath et al. first proposed the idea of modeling the inverse covariance matrix with a sparse structure considering the computing advantage that sparse precision matrix may bring [105]. Unfortunately, the literature [105] does not give results related to the sparseness of the inverse covariance matrix. Based on the conditional relationship formed by two random variables, Bilmes et al. heuristically select the position of zero in the inverse covariance matrix. The results show that only about 70% of the parameters of the full covariance system are needed to achieve the same performance [106]. Hong Kong University of Science and Technology proposed a method to automatically learn the sparse structure of inverse covariance matrix under the HMM framework in order to solve the problem that acoustic model training using full covariance matrix will overfit when training data are insufficient. The specific implementation method is to add the L1 regularization term to the traditional maximum likelihood estimation objective function. The whole training process is as simple as using the maximum likelihood estimation to train the full covariance model. Therefore, a sparse inverse covariance matrix can be used with little additional calculation or programming cost. In addition, since the sparse inverse covariance matrix has clearly modeled the correlation of random variables of feature vectors, there is no need for any de-correlation transformation (for example, semi-tied Covariance Matrix [107]), which simplifies the training process and decoding process. Experimental results on the Wall Street Journal data show that when only about 14 h of training data are available, the WER of the proposed model decreases from 10.5% to 8.77% compared with the full covariance model. Experimental results on Cantonese data from the parliamentary speeches downloaded from the website of the Legislative Council of Hong Kong show that when only about 16 h of training data are available, the WER of the proposed model decreases from 22.87% to 21.02% compared with the full covariance model [83].

On the other hand, in reducing the number of parameters, Povey et al. proposed an acoustic modeling method for SGMM [84]. Although each state in SGMM is also a superposition of multiple Gaussian components, the difference from traditional GMM is that the mean vector and covariance matrix of SGMM mixed Gaussian are not independent parameters but are mapped from the global Gaussian through a vector. Under the same speech recognition conditions, the system parameter scale under the SGMM framework is greatly reduced. Additionally, due to the reduction of the parameter scale, the SGMM model can better overcome the problem of data sparseness in the low-resource environment, so it is more robust. Moreover, since most of the parameter space of SGMM is shared among various states, the only data can be more fully utilized. Under the framework of SGMM, scholars have conducted a lot of research on multilingual training. Burget et al. use different phoneme sets for different languages, while the model parameters of SGMM are shared among multiple languages [85]. The experimental results show that using the shared parameters of the multi-language training acoustic model in SGMM can improve WER. At the same time, SGMM shared parameters are independent of language, and the experimental results do not change with language changes. Therefore, the universality of SGMM enables the parameters learned in resource-rich languages to be successfully reused in resource-limited languages [85]. Based on SGMM, a data borrowing method was born, that is, a method of training SGMM with non-target languages. Microsoft Research has proposed two data borrowing methods combined with SGMM: one method is to minimize KL distance as the objective function of SGMM training, and the other is to optimize KL distance during training and consider state counting optimization criteria [86].

Cross-language acoustic modeling is also one of the methods suitable for low-resource speech recognition acoustic modeling. During the experiment, one or more rich-resource source languages can be used to fully train the acoustic model, and then the target language with insufficient resources can be used to fine-tune the trained acoustic model. The acoustic model trained in this way is more robust, which can greatly alleviate the over-fitting of the acoustic model caused by insufficient target training data. In the existing study, the cross-language SGMM modeling method proposed by Ghoshal et al. proves that the global parameters of SGMM are transferable between different languages under the condition that multiple languages share the same phoneme set, especially when multi-language training parameters are used. Therefore, the acoustic model of the target language with limited resources can be trained by borrowing SGMM global parameters from one or more source languages [15]. In addition, Miao et al. also proposed a cross-language acoustic modeling Subspace Mixture Model (SMM), which is an extension of SGMM. For the target language, SMM learns its subspace parameters through the linear combination of subspaces of the source language. While maintaining the flexibility of SGMM multilingual modeling, SMM reduces the number of parameters required for modeling, making it suitable for target languages with insufficient resources [87].

Next, the main parameters of GMM and DNN are summarized in the acoustic model section. The main parameters of GMM are the mean and variance of each Gaussian component and the weight of each Gaussian component. For a general speech-recognition system, it is assumed that the dimension of the acoustic feature is D, the number of states of the GMM model after the state binding of the decision tree is J, and each state has I Gaussian components. The model parameters of GMM and their numbers are shown in Table 4.

DNN can be regarded as a connection of several layers. The input features are propagated between different layers within the neural network along these connections. Each propagation can be regarded as distorting and deforming the features in a high-dimensional space to obtain new features. For better differentiation, each layer has a different internal structure. Each layer includes several neurons, whose function is to stimulate the output of each neuron according to the input characteristics and provide it to the subsequent layers. In speech recognition, acoustic features are natural high-dimensional signals, so their feature dimensions correspond to the number of neurons in the input layer of the neural network. We refer to the input of the DNN as the input layer, the output as the output layer, and the layers in between as the hidden layer. DNN consists of an input layer, multiple hidden layers, and an output layer. Each layer has fixed nodes, and nodes between adjacent layers are fully connected. Each output neuron of a fully connected node is a linear transformation of all its input features, the output neurons are independent of each other, and the coefficients of the linear transformation are the parameters of the node. The neural network described here simulates the neurons in the human brain, which are connected to each other and transmit signals to the next neuron through axons. An axon can scale the size of the signal passing through it, and this scaling factor is called the weight. As shown in Figure 3, it is a DNN with a fully connected hidden layer and a fully connected output layer. The two fully connected layers have five and four neurons, respectively, and the weights between different neurons in the same layer are different. For example, if the fully connected hidden layer has four neurons and the input layer has three neurons, then each hidden layer neuron has three connection weights corresponding to the input layer, and the input weight of the entire layer can be represented by a matrix of 4×3.

Equation (4) shows the mapping of input value $\left\{x_{1},x_{2},x_{3},\ldots ,x_{j}\right\}$ to output value $y_{i}$, where $w_{ji}$ is the weight of each connection, b is the offset, and the output of the neuron $z_{i}=\sum_{j}w_{ji}x_{j}+b$. $z_{i}$ further output through the activation function $y_{i}$:(4)$$y_{i}=f\left(z_{i}\right)$$
where $f\left(\cdot \right)$ is the activation function.

For DNN, the output of each hidden layer must have an activation function, otherwise it makes no difference whether there is a hidden layer or not. Common activation functions are Sigmoid, Tanh, and ReLU, as shown in Table 5 below.

Among them, the sigmoid function is monotonically increasing, continuously derivable, and its derivative form is very simple, so it is a suitable activation function and has a wide range of applications in DNN. However, it also has three disadvantages: one is that its calculation includes function operations, and division is required in the derivation of the error gradient, which makes the calculation very complicated; the second is that the gradient disappears due to oversaturation (the input value is too large or is too small, with the problem being that the gradient approaches 0), so the training of DNN cannot be realized; the third is that the output of the function is not centered at 0, so the upper layer receives data that are not centered at 0, resulting in the gradient falling wobbly, which makes weight update less efficient. The Tanh function can solve the above three problems, and its output is 0-centered. The main advantage of the ReLU function is linear non-saturation, which has a faster convergence rate than the Sigmoid/Tanh function in gradient descent. In addition, it only needs to judge whether it is greater than 0, so the calculation speed is also very fast. There are two disadvantages: one is that the output is not centered at 0; the other is that when the output is negative, the ReLU function will not be activated, and the corresponding parameters will not be updated.

When the DNN is combined with the HMM, the Layout output nodes of the DNN correspond one-to-one with the state nodes of the HMM. So, the observation probability of each state can be obtained through the output of DNN. The transition probabilities and initial state probabilities of the HMM remain unchanged. The estimation of posterior probability in HMM by DNN does not require very strict data distribution assumptions, especially, the feature elements of the same frame do not need to be independent of each other. Therefore, the acoustic features do not necessarily need to use MFCC or PLP, but more primitive features such as Fbank can be used. As mentioned above, the parameters of the DNN include the layers and the number of nodes (neurons) in each layer. Layers include weight matrix (input weight of input layer, hidden weight of hidden layer, and output weight of output layer) and offset (offset of input layer, offset of hidden layer, and offset of output layer). Of course, the parameters of the DNN also include the activation function. For a general speech-recognition system, it is assumed that the dimension of the input feature Fbank is A, the hidden layer has B layers, and each layer has C nodes. The output layer nodes should correspond to the HMM states one-to-one (assuming there are D HMMs in total and each HMM has E states). The model parameters of DNN and their numbers are shown in Table 6.

## 5. Low-Resource Speech Recognition Resource Expansion

The resources in speech recognition consist of data resources and pronunciation dictionaries. Data resources include speech audio and corresponding annotation text. Expanding data resources is a very important method in low-resource speech recognition. It includes two methods. One is to generate corresponding audio for existing text, and the other is to generate corresponding labeled text for existing audio. The expansion of pronunciation dictionary is based on Grapheme-to-Phoneme (G2P) conversion and web crawling.

### 5.1. Data Resource Expansion

Due to insufficient data resources in a low-resource environment, the model training is insufficient and the desired performance cannot be achieved. As a measure to solve this shortcoming, training data expansion shows superior performance. Its purpose is to find audio and text resources with the correct correspondence to join the training set. The research done by related researchers is shown in Table 7 below. Among them, the corresponding audio generated by the existing text is mainly embodied on the basis of the existing training data, keeping the semantics unchanged (that is, the labeled text unchanged) to transform or process the audio and change its channel, speech rate, and other characteristics.

At present, the commonly used methods in research include Vocal Tract Length Perturbation (VTLP), noise addition, and Speed Perturbation (SP). Specifically, VTLP generates a data enhancement scheme for new samples by perturbing or distorting the speech spectrum of existing training samples, linearly stretching or compressing the speech signal in the frequency domain, and changing the speaker’s vocal tract length to achieve data enhancement [108,109]. Adding noise is one of the classic data expansion methods. Data enhancement is achieved by adding different types of noise with different signal-to-noise ratios to the speech [110,111]. SP is to generate a regular time signal, which essentially stretches or compresses the speech signal linearly in the time domain. In simple terms, the speaker’s speech rate changes in the same sentence to achieve data enhancement [112].

Speech synthesis is also a form of data augmentation that converts existing text into speech. When there is very little or no existing text, the data are augmented with zero-resource speech synthesis. Zero-resource speech synthesis is the task of building text-to-speech (TTS) models without transcription. Karthik et al. participated in the Zero Speech Challenge 2020 (Zerospeech 2020). The research content of Karthik et al. includes two methods: (1) propose a flow chart for generating an acoustic unit (DAU) and (2) propose a synthesis system of 4 TTSs. In this study, speech was modeled as a series of transient and steady-state acoustic units, and a unique set of acoustic units was discovered through iterative training. The Discovery Acoustic Unit (DAU) process for generating acoustic units is shown in Figure 15. First, speech is segmented into syllable-like units. The similarity matrix is obtained by computing the DTW scores between all pairs of syllable class segments. Homogeneous syllable class units are clustered using the K-Nearest Neighbor (KNN) graph clustering method. The syllable unit in each cluster is a sequence of 3 AUs, corresponding to rising transients, steady-state, and falling transients. HMM is used to model AU. The syllable-like units are transcribed using the trained model. The obtained transcriptions are then used to retrain the model. The training process and transcription process are repeated until convergence. This process of repeated training and transcription is called self-training. The initial model thus obtained is trained only on syllable-like fragments present in the cluster. The self-training process of the syllable segment is called the first stage training. Using the initial model, complete syllable-like fragments are transcribed. Train yourself on this ensemble to get a better model. Once the models are trained on the entire set of syllable-like segments, the models are used to transcribe continuous speech. Phase 2 self-training is performed on continuous speech until convergence. The model thus obtained is the final model used to generate the AU sequence. The end-to-end speech synthesis framework is only given speech waveforms and corresponding text transcriptions (in this case sequences of acoustic units) to train synthesizers. It alleviates the need for separate modules for feature engineering and language-specific tasks. The TTS framework used in this work has two phases. The first stage maps the AU symbol sequence to the corresponding spectrogram. The second stage inverts the spectrogram back to speech. The four synthetic systems proposed in this study are shown in Figure 16. For System 1, the end-to-end framework used is based on the Tacotron2 architecture. Tacotron2 consists of an encoder and a decoder with attention weights. It is responsible for converting the AU sequence to a mel spectrogram. The encoder extracts sequence information from character embeddings, and the attention module predicts a fixed-length context vector. The decoder predicts a frame-level mel-spectrogram at each step. For speech waveform inversion, the WaveGlow vocoder is used [124]. WaveGlow takes a mel-spectrogram as input and generates speech output. Since the task is to synthesize speech in the target speaker’s voice, the inversion phase is fixed. Three different methods are proposed to efficiently learn the mapping between AU sequences and spectrograms. The proposed method explores an end-to-end framework including speaker embedding (System 2), hierarchical training (System 3), and gender-dependent training (System 4), to learn mappings. End-to-end TTS is trained based on the Tacotron2 architecture. The x-vectors are used as speaker embeddings to generate speech in the target speaker’s voice. In hierarchical learning, similar to AUD methods, the mapping is first limited to smaller units. The obtained model is then used to guide the learning of complete utterances. Since the training data has both male and female speakers, a gender-dependent TTS is developed. During synthesis, an appropriate TTS is employed according to the gender of the target speaker. It is important to note that the synthesis pipeline of System 1 is common to all the other three proposed systems. The final systems Karth et al. submitted to the competition committee were System 1 and System 2 (the x-vector model). There are two languages used for experiments: English and Indonesian. According to the experimental results, the overall synthesis quality of system 2 (x-vector model) is better than that of system 1. The improvement in character error rate is significant when System 2 (the x-vector model) is used for spectrogram mapping. There is absolute improvement in CER character error rate of 16% (0.61→0.45) for English and 23% (0.67→0.44) for Indonesian [113]. The Primate Research Institute of Kyoto University in Japan also participated in Zerospeech2020 and proposed its own zero-resource speech synthesis system. The research content of the Primate Research Institute of Kyoto University, Japan mainly includes two parts: (1) treat the unsupervised learning of human language as a biological/psychological problem and use biological/psychologically motivated ANN modules to deal with the challenge; (2) the proposed TTS model consists of three modules. The model proposed by the research team consists of an auditory module (encoder), a symbolic module (ABCD-VAE, a novel discrete VAE whose first four letters represent attention-based classification sampling of the Dirichlet prior), and a pronunciation module (decoder). The auditory module processes MFCC frames using an echo state network (ESN) and simulates computations in a cortical microcircuit. The output of the auditory module is a time series of real-valued vectors, and the role of the symbolic module is to discretize them. Ideally, the module should classify sound frames into linguistically meaningful categories, such as phonemes where utterances span frames. Since the size of phonemes in the inventory varies from language to language, the symbol module should also detect an appropriate number of classes rather than just putting frames into a predetermined number of classes. A popular approach to such clustering problems in computational linguistics and cognitive science is Bayesian clustering based on the Dirichlet distribution/process. The pronunciation module receives the output of the symbol module and generates waveforms from it. The pronunciation module in this study adopts the neural network implementation of the Neural Source-Filter Model. The experiments were conducted in two languages: English and surprise. The submitted system achieves better recognition rates than the baseline. For English, the relative improvement in CER was 10% (0.77→0.67). For surprise, the relative improvement in CER was 21% (0.67→0.46). Experimental results show that the encoder extracts important linguistic features, while the decoder recovers them in speech synthesis [114].

Another way to expand data resources, that is, to generate corresponding labeled text for existing audio, mainly includes semi-supervised training [115]. A small number of existing fine labeled data are used to train the seed model, the unlabeled data are decoded by the seed model to obtain hypothesis annotation, and then the hypothesis annotation data with high reliability is selected through certain data screening strategies (e.g., confidence degree, confusion degree, etc.). Finally, the model is retrained by mixing the selected hypothesis annotation data and the original fine labeled data to obtain the final acoustic model.

It can be seen from Table 6 that most of the research on data enhancement focuses on a single data enhancement scheme and rarely considers the complementarity between different data enhancement technologies. Cui et al. studied the complementary effects of VTLP and Stochastic Feature Mapping (SFM). SFM does not rely on frequency spectrum changes in the specified feature space and realizes statistical voice conversion between speakers, that is, statistically converting one speaker’s voice data into another speaker’s voice data to enhance training samples. When the two enhancement techniques are used in an acoustic model at the same time, the recognition rate is reduced [116]. The research team of Cambridge University firstly studied the single enhancement effect of VTLP and semi-supervised training and then studied the complementary effect of semi-supervised training and VTLP [117]. Experimental demonstration has been carried out on two limited language packages (Assamese and Zulu) published by IARPA Babel. The specific architecture adopted by the speech recognizer and the nature and quantity of the enhanced data used determine the utilization of the data. The team used these data to conduct experiments in a tandem architecture. The tandem structure used is shown in Figure 17, where the MLP has an input layer, a hidden layer, and a bottleneck layer. In the experiment, PLP standard is parameterized, optionally de-correlated and transformed, sent to the input layer, and then reaches the bottleneck layer through a series of nonlinear transformations of the hidden layer. MLP extracts BN features, connects them with the initial features, and uses them in the GMM-HMM speech recognizer. Specifically, the researchers used a consistent process to create a tandem system: first build GMM-HMM independent of the speaker based on PLP, then train MLP, and finally build MLP and GMM in tandem. The team did four sets of comparative experiments on the two languages: (1) a baseline system that does not use any enhancement technology; (2) a system that uses a single VTLP enhancement; (3) a system that uses a single semi-supervised training enhancement; and (4) uses VTLP and semi-supervised training joint enhanced system. Through the experimental evaluation of the token error rate (TER) in percentage, the results show that (1) for Zulu, the TER of the baseline system without any enhancement technology is 78.4%; the TER of the system enhanced by VTLP alone is 77.1%; the TER of the system enhanced by semi-supervised training alone is 77.7%; and the TER of the system enhanced by VTLP and semi-supervised training is 76.7%. (2) For Assamese, the TER of the baseline system without any enhancement technology is 69.4%; the TER of the system enhanced by VTLP alone is 69.3%; the TER of the system enhanced by semi-supervised training alone is 67.6%; and the TER of the system enhanced by VTLP and semi-supervised training is 68.3%. From the experiments of this team, it can be concluded that whether it is VTLP alone or semi-supervised training alone, the performance of low-resource languages Assamese and Zulu systems can be improved, but the combined scheme only produces gains for Zulu, and the combined effect of Assamese is not as good as using semi-supervised training alone.

### 5.2. Pronunciation Dictionary Extension

As one of the important resources in the speech-recognition system, the pronunciation dictionary is an important bridge between the acoustic model and the language model, which determines the recognition vocabulary of the speech-recognition system. The pronunciation dictionary should cover all the words encountered in the recognition process as much as possible to ensure the good recognition performance of the system. Otherwise, the problem of out-of-vocabulary will occur, which will seriously reduce the recognition rate of the system. Currently, commonly used dictionary expansion strategies are divided into two categories: one dictionary expansion strategy is through web crawling, and the other is based on G2P conversion.

Dictionary expansion through web crawling is to directly search and crawl the correct pronunciation of words to complete the dictionary expansion. The shortcoming of the application of this scheme is that it requires a lot of research on the language. Researchers from Karlsruhe Institute of Technology proposed to use network-derived pronunciation to build a pronunciation dictionary from scratch. This method can be implemented only by extracting pronunciation and corresponding written words from the World Wide Web [118]; specifically, the researchers used an automatic dictionary extraction tool (part of the Rapid Language Adaptation Toolkit (RLAT)), which took a vocabulary as input and looked for matching pages in an online dictionary. The work done by the team was mainly divided into three aspects. (1) Wiktionary was analyzed from the perspective of language and vocabulary coverage and compare it with the multilingual pronunciation dictionary GlobalPhone in terms of quality and coverage. In addition, two quality checks were performed: first, the pronunciation in Wiktionary was compared with the pronunciation in the GlobalPhone dictionary. Second, the influence of Wiktionary Dictionary pronunciation as a pronunciation variant on the ASR system was analyzed. (2) The GlobalPhone and Wiktionary’s word-pronunciation pairs created statistical G2P models for European languages. (3) A fully automatic method was proposed to detect, delete, and replace inconsistent or defective entries in Wiktionary Dictionary word-pronunciation pairs [118]. The experiment found that in the process of crawling the text in the target domain of the ASR system and automatically performing text normalization, extracting the vocabulary of the normalized crawling text, and creating a pronunciation dictionary with the vocabulary based on the pronunciation of the World Wide Web, new field or new language dictionary can be generated without humans, effectively reducing complexity and human error. However, a quantitative check of the list of international cities and countries shows that even in the phonetic system of a language, it may not be possible to find a proper name for the pronunciation. Among the languages studied by the team, about 15,000 phoneme markers are enough to train a stable G2P model. Experiments have proved that the rapid growth of languages in Wiktionary can effectively alleviate the problem of language pronunciation with insufficient resources.

The pronunciation dictionary extension based on G2P conversion discovers new words by counting a large number of text corpora and uses G2P conversion to obtain the pronunciation of new words and add them to the dictionary to complete the dictionary extension [119,120,121,125]. Researchers present a novel data-driven language independent approach for grapheme to phoneme conversion, which achieves a phoneme error rate of 3.68% and a pronunciation error rate of 17.13% for English. They apply their stochastic model to the task of dictionary verification and conclude that it is able to detect spurious entries, which can then be examined and corrected by a human expert [126]. The core of this method is G2P conversion. Only by constructing a high-performance G2P converter can the correct pronunciation of words be obtained. However, the performance of G2P pronunciation very much depends on the similarity between the words and the vocabulary in the dictionary used by the training model. Therefore, this method is prone to conversion errors for words with large spelling differences. In order to provide G2P with “reliable” pronunciation, Schlippe et al. proposed a two-stage filtering method to detect, replace, and delete inconsistent or defective entries in the dictionary. The two-stage filtering process is shown in Figure 18 [122]. In the first stage, word-pronunciation pairs are pre-filtered by applying three first-level methods to filter word pairs by length filtering (Len), Epsilon filtering (Eps), or M-N alignment filtering (M2NAlign). Specifically, (1) Len means that if the ratio of the glyph to the phoneme mark exceeds a certain threshold, the pronunciation is deleted. (2) Eps first executes 1–1 G2P alignment, which includes inserting glyphs and phoneme nulls (epsilons) into the lexical entries of the word and judging whether the ratio of glyphs and phoneme nulls exceeds the threshold. (3) M2NAlign performs M-N G2P alignment first and then deletes pronunciation when the alignment score exceeds the threshold. The threshold for each filter condition depends on the mean μ and standard deviation σ of the focus measurement (calculated on all word-pronunciation pairs), that is, the ratio between the number of glyphs and phoneme marks in Len, the number of glyphs and phoneme nulls in Eps Ratio, and the alignment score in M2NAlign. Those word-pronunciation pairs with result numbers shorter than μ∗σ or longer than μ+σ are rejected. The second stage is to use G2P on the remaining word pronunciation pairs. These two-stage filtering methods are denoted as G2PLen, G2PEps, and G2PM2NAlign in the following experimental part. The team verified the performance of the proposed two-stage filtration system through three different dictionaries: (1) word-pronunciation pairs in Czech, English, French, German, Polish and Spanish Wiktionary; (2) GlobalPhone Hausa pronunciation dictionary; and (3) new pronunciation variants to supplement Mandarin–English SEAME code conversion dictionary [127,128,129]. In terms of Wiktionary, the experimental results show that the WER of Eps, M2NAlign, G2PLen, and G2PM2NAlign are relatively reduced by 27.3%. However, the success rate of each method varies from language to language. The biggest improvement of this study is the 12.25% reduction in WER on the English Wik dictionary word-pronunciation pair using M2NAlign. As for the GlobalPhone Hausa dictionary, the experimental results show that, except for G2P, the WER increased by 0.19% (23.49%→23.68%), and the WER of all the remaining filter dictionaries decreased. Among them, G2PLen has the best performance, with a relative decrease of 0.61% (23.49%→22.88%). In the case of SEAM, after using the filtered new pronunciation for decoding, compared with existing dictionaries, the two-stage filtering can reduce the MER by 0.2% [122].

In addition, the application of acoustic properties to the generation of pronunciation dictionaries can be considered as an extension of the G2P method [123,130,131]. In these methods, acoustic data are used to construct the relationship between phonemes and glyphs, which can directly infer the phoneme sequence of new words, or to estimate the weight of the new word alternative pronunciation generated by the aforementioned G2P converter. The G2P converter proposed by the Speech Technology Research Centre at the University of Edinburgh is trained by a small seed dictionary to obtain multiple speeches of new words, which are then re-scored using viterbi approximation and expectation maximization algorithm based on weighted finite state sensors [123].

## 6. Technical Challenges and Prospects

Compared with speech recognition technology with a large amount of data resources, the performance of low-resource speech-recognition systems is obviously at a disadvantage. The three major technical methods of feature extraction, acoustic model, and resource expansion are all facing difficulties and urgently need to be overcome.

In terms of feature extraction, common traditional shallow features such as MFCC and PLP are not robust enough. In a low-resource environment, the requirements of system modeling cannot be met. In order to obtain more robust feature parameters, traditional shallow features are usually non-linearly transformed to extract deep features. Considering the acoustic similarity between different languages, rich-resource source language data can be used to train the target language, reducing the requirement on the amount of target language data, thereby solving the problem of the target language data amount. In addition, different posterior features or acoustic features can be spliced and fused to enhance feature recognition performance. Because of its own advantages, neural network technology has been effective in the application of low-resource speech recognition feature extraction. However, the requirement of neural network training for large data volume and the current situation of insufficient data volume in low-resource environments are still a contradiction. Knowing how to use limited data resources to train neural networks is still a problem that needs continuous research.

In terms of acoustic modeling, the traditional speech-recognition system constructed by training GMM-HMM through the underlying acoustic features has gradually been unable to meet the requirements of low-resource speech recognition tasks. The existing Tandem system is mainly modeled by BN features. Aiming at the shortcoming that the BN layer reduces the accuracy of DNN classification when extracting BN features in the Tandem system, a high-level feature extraction method that does not change the DNN training structure can be used for optimization. The specific implementation methods are as follows: first train a DNN that does not contain the BN layer, and then use the non-negative matrix factorization algorithm to decompose the hidden layer weight matrix to obtain the base matrix as the newly formed feature layer. When the offset vector is not set in this layer, the linear output of this layer is used as a new low-dimensional high-level feature through data forward propagation.

Data sparsity is one of the most challenging problems in the construction of DNN-based acoustic models. The current solutions are divided into two types: (1) use sparse data to build a simpler neural network and (2) build a deeper and wider network by sharing data and model parameters from high-resource data sets. For the multi-language training method of DNN, the improvement of the balance of the training process is not enough, that is, when the same method is applied to low-resource conditions, it is easy to produce language bias, which causes unbalanced training and leads to overfitting. Only by mapping the introduced multi-language speech data to a relatively similar distribution space can the auxiliary training be made more effective. Aiming at the problem of a sharp decline in the acoustic modeling performance of multilingual DNN features under the condition of low-resource training data, the following specific methods can be used to optimize. In the auxiliary training of BN-DNN, in view of the feature that the BN layer is located in the shared layer, technologies such as dropout, maxout, and ReLU are introduced to improve the over-fitting problem caused by the irregular distribution of multi-stream training samples.

In addition, the modeling method of SGMM needs to be improved. Although SGMM is suitable for low-resource speech recognition, it is not suitable for multilingual training. Therefore, we can consider introducing multi-language training into the feature extraction process of DNN and then train SGMM to construct the Tandem system. In recent years of speech recognition research, CNN has shown better performance than DNN. However, under low-resource conditions with insufficient training data, there is a problem that network parameters cannot be fully trained, and traditional multi-feature stitching cannot be used. The method improves the performance of the model. At this time, optimization is performed by constructing a parallel convolution sub-network based on the multi-type features of the low-resource training data.

Due to the limitation of data resources in low-resource environments, speech-recognition systems have higher requirements for acoustic models. The speech recognition acoustic model in low-resource environments should have more powerful information utilization capabilities and make full use of information in limited data. In addition to CNN, some other neural networks have emerged in recent years, such as long- and short-term memory loop neural networks, Connectionist Temporal Classification, multi-dimensional RNN, and hierarchical down-sampling RNN. These networks can make full use of the network structure of rich information in speech signal and effectively improve the performance of low-resource speech recognition.

In resource expansion, it can be seen from Table 6 that data resource expansion is divided into two types. One is to generate corresponding audio for existing text. The main methods are VTLP, noise addition, SP, etc. The other is to generate corresponding labeled text for existing audio. The main method is semi-supervised training. At present, most experiments use a single training data expansion method, and the superimposed use of multiple expansion methods faces the possibility of increased confusion of the data set and reduced systemicity. The automatic pronunciation dictionary generation method has superior performance for generating the pronunciation of new words in some specific languages, but it must be based on obtaining a large amount of text corpus and filtering out new words that are not in the original vocabulary, which greatly increases the complexity of the algorithm. In addition, the acquired text corpus has a certain degree of randomness and one-sidedness. The number of out-of-vocabulary covered in the corpus is unpredictable, and most of the words in the corpus are known words, and only a few of them are out-of-vocabulary. Therefore, the automatic generation method of pronunciation dictionary needs to be improved, and the correctness of the crawled text can be compared and improved by crawling more different websites. 

In recent years, due to the rise of the E2E model, people have applied the E2E model to speech recognition technology. Traditional speech recognition (GMM-HMM/DNN-HMM) has many assumptions at the theoretical level. The modular design is not easy to deploy programs in hardware, and it cannot fully meet the ultimate needs of human-machine voice communication. Additionally, end-to-end speech recognition technology can directly complete speech-to-text transcription through a single model. While the current E2E speech recognition technology has great potential for development, there are also many problems. E2E speech recognition technology is data-driven training. The performance of the model depends largely on the number of available labeled corpora. E2E speech recognition requires more training data than traditional speech recognition. The main reason is that the parameter scale of a single ensemble model is very large, which can easily lead to overfitting of the model. Low-resource E2E speech recognition needs to be researched, and the E2E speech recognition rate needs to be improved through better modeling methods without acquiring additional target training data. Traditional speech recognition technology has better recognition performance in some low-resource speech recognition tasks, while E2E speech recognition technology is often more advantageous when there is sufficient training data. The main reason is that the training of the existing end-to-end model is data-driven, the parameters in the model are updated through the calculation of the gradient, and the calculation of the gradient is easily affected by the model training mechanism and the information expression ability of the data. There is no expert knowledge involved in the training process, so the training of the model lacks certain knowledge guidance and has a certain blindness. It is generally considered that it is difficult to use the existing E2E speech-recognition models to obtain good optimization and sufficient training when the training data are limited. Therefore, it is hoped that researchers can solve the low-resource E2E speech recognition problem by reducing model complexity and improving learning mechanism, thus ushering in a new era of E2E low-resource speech recognition.

## 7. Discussion

This paper is devoted to the study of speech recognition in low-resource situations and mainly discusses the current technology from three aspects: feature extraction, acoustic model and resource expansion. In the feature extraction module, two parts are mainly described: general feature extraction technology and DNN-based feature extraction technology. In the general feature extraction technology section, the steps of extracting MFCC, Fbank, and PLP are introduced in detail. At the same time, the method of feature transformation is also introduced in this part. In the part of extracting features based on DNN, in addition to introducing the deep features themselves, it also introduces the robustness of features through splicing. In the acoustic model module, three parts are mainly described: establishment of general acoustic model, establishment of acoustic model based on DNN, and establishment of acoustic model based on improvement of GMM. In the general acoustic module part, due to the short-term stationary characteristics of speech signals, HMM is used for modeling. So, at present, acoustic modeling is mainly divided into two types: GMM-HMM and DNN-HMM. In addition, DNN has different structures, including CNN, LSTM, TDNN, and TDNN-F, etc. One point to emphasize is that the HMM architecture of DNN-HMM is generic. However, DNN can use different network models such as CNN, LSTM, TDNN, and TDNN-F. In the resource extension part, two parts are mainly described: the extension of the data resource and the extension of the dictionary resource. In the extension part of the data resource, it mainly includes two parts: extension through existing text and extension through existing speech. Extensions with existing text include VTLP, adding noise, SP, and speech synthesis. There is semi-supervised training as a way to extend through existing speech. The extended dictionary resources are mainly expanded through the pronunciation dictionary expansion based on G2P conversion and the pronunciation dictionary expansion through web crawling. For the resource extension part, there are no special requirements, and data enhancement can be performed normally. Next, I summarize which acoustic features are used, how these features should be transformed and when to be concatenated, and which acoustic models are used; the combined use of these factors is listed in Table 8 below.

The paper mainly introduces the content of three parts: feature extraction, acoustic model, and resource extension. Here, the more classic methods are listed in Table 9 below. The content contained in the table mainly covers the first two parts: feature extraction and acoustic model, and the best WER or WER improvement. For the resource extension part, we will not list its extension methods in the table. Because in the data extension part (mentioned VTLP, adding noise, SP, and speech synthesis), no matter which extension method is used, it can achieve the purpose of data enhancement. However, when several are used in combination, special attention is required to achieve the purpose of data enhancement. Only when SP is combined with other methods can the purpose of data enhancement be achieved, and other combined methods may not necessarily achieve complementary effects. In the dictionary expansion part, there are two ways to expand, mainly through the expansion of pronunciation dictionary based on G2P conversion and the expansion of pronunciation dictionary through web crawling. Moreover, the expansion of the dictionary is a separate part. As long as the interference of words outside the set is excluded, the accuracy of recognition will be improved by expanding the capacity of the dictionary.

As described in Table 9, we list the classical methods in it. Common features are MFCC, PLP, and Fbank. When training the GMM-HMM acoustic model, due to the limitation of computation, the diagonal covariance matrix is usually used, so the dimensions of the GMM probability density function are conditionally independent, so the MFCC feature is usually used. When training the DNN-HMM acoustic model, because the DNN does not require very strict data distribution assumptions for the estimation of the posterior probability in the HMM, especially the feature elements of the same frame do not need to be independent of each other, so the acoustic features can use more primitive features like Fbank. The HMM architecture of DNN-HMM is generic. Among them, DNN can use different network models, such as CNN, LSTM, TDNN, and TDNN-F. The following compares the advantages and disadvantages of these common DNN structures. In addition to context-related information, speech signals also contain various frequency features, which are different between frames and within each frame. These local differences are not well captured by traditional modeling methods. The speech signal can be represented by a spectrogram, so the local features can be extracted on the time axis and frequency axis by referring to the method of image processing. CNN simulates speech as a two-dimensional “image” and extracts local features in the time and frequency domains through a relatively small perception field of view. In 1997, the structure of LSTM was introduced into RNN. LSTM successfully overcomes the fundamental problems of traditional RNNs and can perform well tasks such as pattern recognition of noisy input sequences that RNNs cannot cope with. Both RNN and LSTM have general computing power, but unlike conventional RNN, LSTM is more suitable for learning from input time series, to obtain the ability to classify, process, and predict the sequence. In 1989, A.Waibel proposed TDNN and confirmed in experiments that TDNN has better performance than HMM. The two main features of TDNN are that it can adapt to dynamic time-domain feature changes and has fewer parameters. The layers of traditional DNN are fully connected. The change of TDNN is that the features of the hidden layer are not only related to the input of the current moment but also related to the input of the past and future moments. The input of each layer of TDNN is obtained through the context window of the lower layer, so it can describe the temporal relationship between nodes in the upper layer. In 2018, the famous phonetic scholar Daniel Povey made further improvements to the TDNN network and proposed a factorized TDNN, that is, TDNN-F with a semi-orthogonal matrix. TDNN-F is based on Singular Value Decomposition (SVD). As the best-known matrix factorization method, SVD is an efficient way to reduce the size of the trained model because with SVD we can decompose each weight matrix into two factor matrices, optimizing the network parameters by discarding relatively smaller singular values. The internal structure of TDNN-F is the same as TDNN compressed by SVD. Although it is more efficient to directly train the model with the bottleneck layer after random initialization of the TDNN compressed by SVD, the problem of unstable training occurs from time to time. To avoid this, TDNN-F constrains one of the two decomposition factor matrices from being a semi-orthogonal matrix after the SVD decomposition. This approach not only meets the requirements and characteristics of SVD but also does not lose the modeling ability of the network. As can be seen from Table 9, these DNNs can be used as acoustic models alone or in combination. In addition to the three common characteristics mentioned at the beginning, there are some remaining characteristics. However, in any case, the features in the table are matched with the respective selected acoustic models to achieve an objective recognition rate. In Table 9, in addition to the datasets, features, and acoustic models used in the experiments, the best WERs in the experiments or the improved WERs of the systems are also listed. In column 4 in Table 9, the header is divided into two parts: one is the best WER, and the other is the improvement of WER. For differentiation, the best WER is denoted as (1) and the improvement in WER as (2).

Furthermore, although this paper presents the current state-of-the-art research in low-resource speech recognition, one point must be made clear. Currently, no single model is perfect and can accommodate all low-resource situations. The same conclusion can be drawn from [82]. Although a certain model gets a good recognition rate in one language, the results differ in another language because different languages represent different datasets. These datasets are diverse in morphological properties, sound systems, writing systems, speech collection strategies, speaker pools, and recording settings. So, no single model is suitable for all low-resource speech recognition.

## 8. Conclusions

The application of speech recognition technology is no longer limited to high-resource languages such as English and Mandarin Chinese. Knowing how to build a high-performance speech-recognition system in a low-resource environment has become an international research hotspot and difficult problem. This paper briefly describes the cutting-edge progress of low-resource speech recognition from three aspects: feature extraction, acoustic model, and resource expansion. It focuses on the analysis of many technical challenges in realizing reliable low-resource speech recognition and looks forward to future work, hoping that the work in this paper can provide help for colleagues who study low-resource speech recognition.

## Figures and Tables

**Figure 1 sensors-23-09096-f001:**
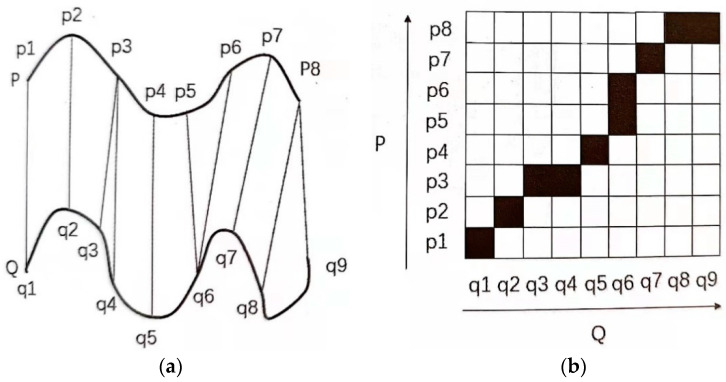
The basic framework of DTW. Figure (**a**) shows the unequal length matching of P and Q speech through time bending. Figure (**b**) transforms the matching similarity problem between speech P and Q into the optimal path problem.

**Figure 2 sensors-23-09096-f002:**
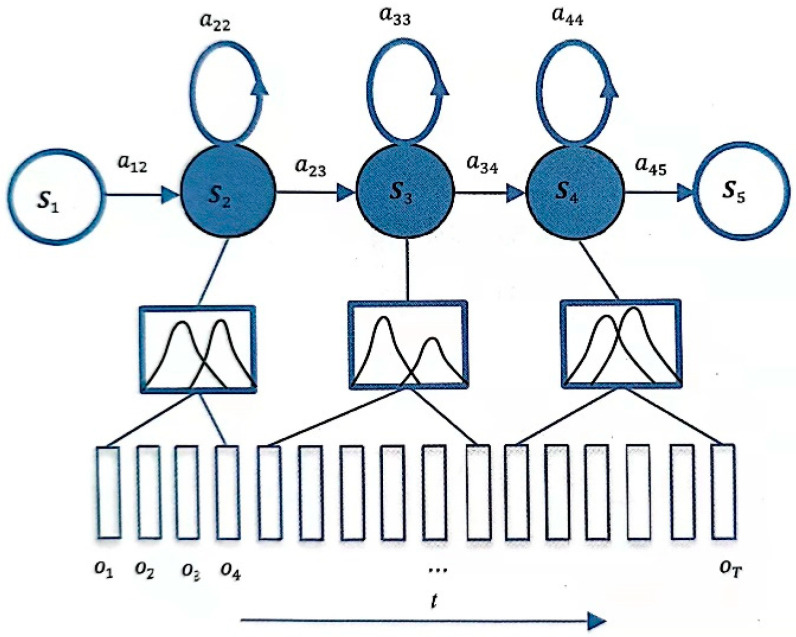
The basic framework of GMM-HMM.

**Figure 3 sensors-23-09096-f003:**
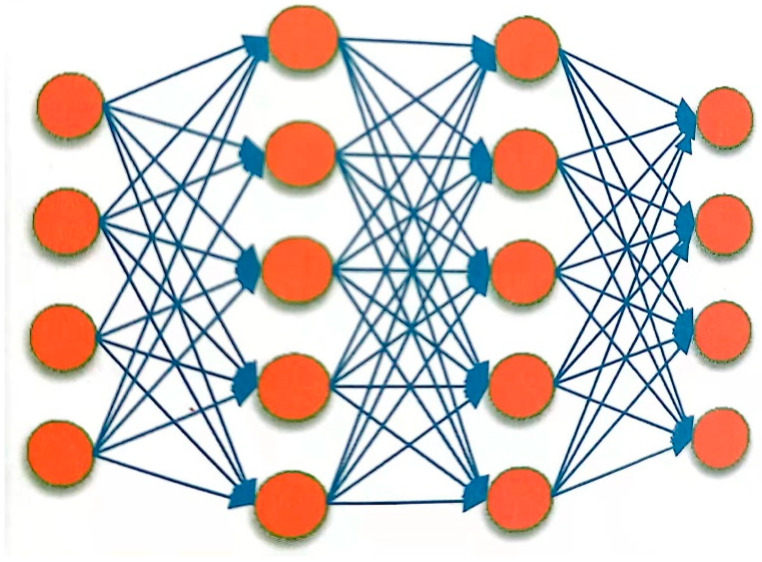
The basic framework of DNN.

**Figure 4 sensors-23-09096-f004:**
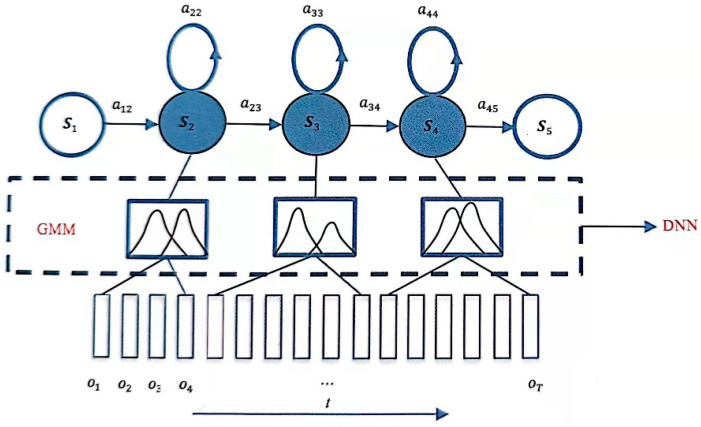
The basic framework of DNN-HMM.

**Figure 5 sensors-23-09096-f005:**
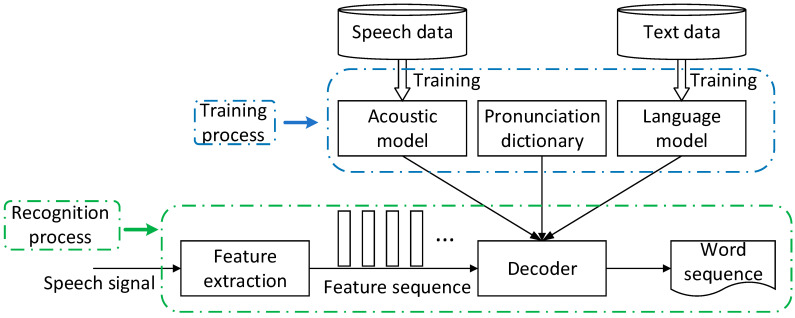
The basic framework of speech recognition corresponding to GMM-HMM/DNN-HMM.

**Figure 6 sensors-23-09096-f006:**
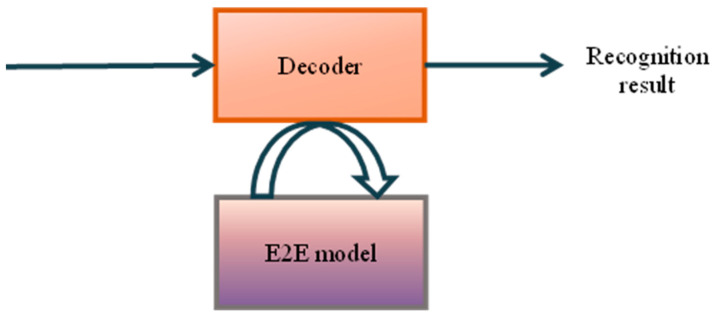
The basic framework of E2E.

**Figure 7 sensors-23-09096-f007:**
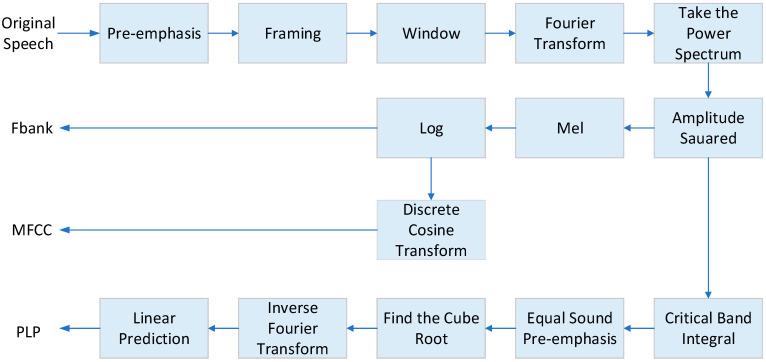
Commonly used acoustic feature extraction process.

**Figure 8 sensors-23-09096-f008:**
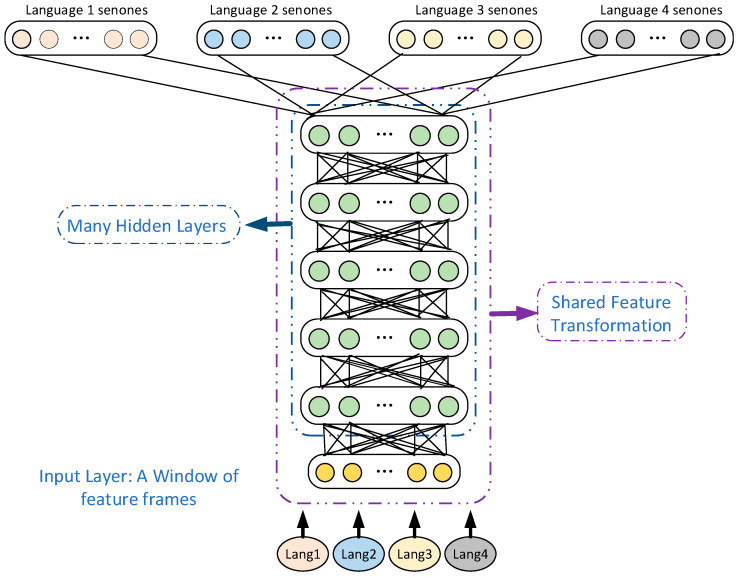
Architecture of shared hidden layer multilingual DNN [44].

**Figure 9 sensors-23-09096-f009:**
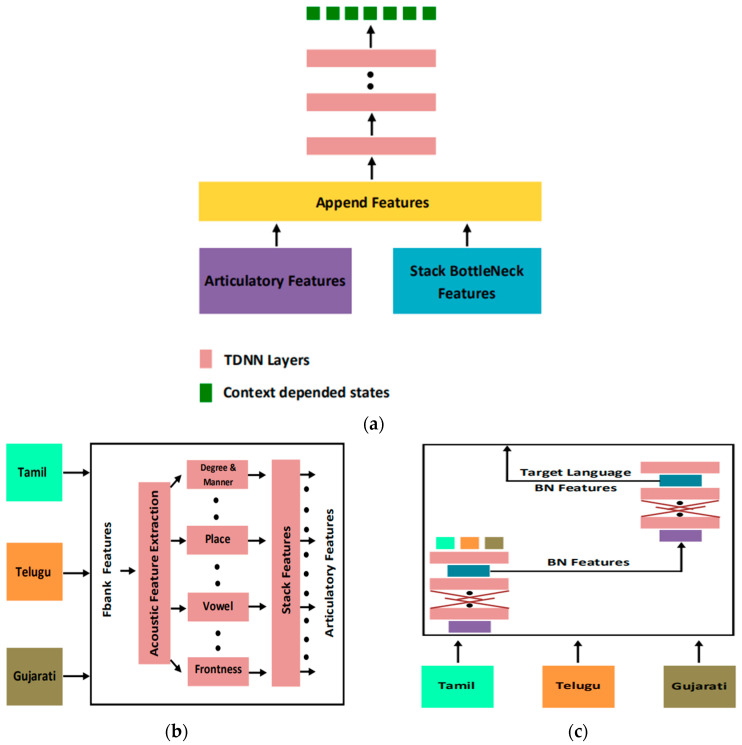
(**a**) The feature extraction model [52]; (**b**) the pronunciation classifier used to extract AF [52]; and (**c**) the feature extractor used to extract SBF [52].

**Figure 10 sensors-23-09096-f010:**
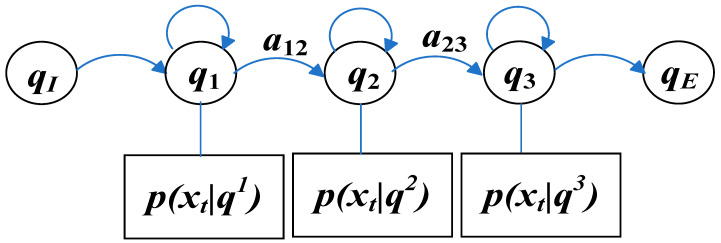
The framework of a three-state HMM.

**Figure 11 sensors-23-09096-f011:**
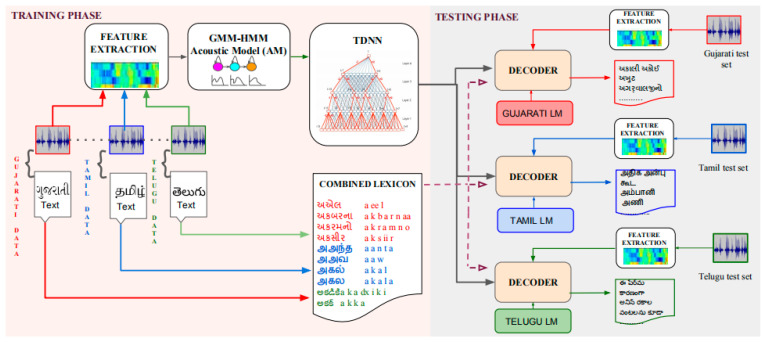
Architecture of ASR system submitted for the IS 2018 Challenge [75].

**Figure 12 sensors-23-09096-f012:**
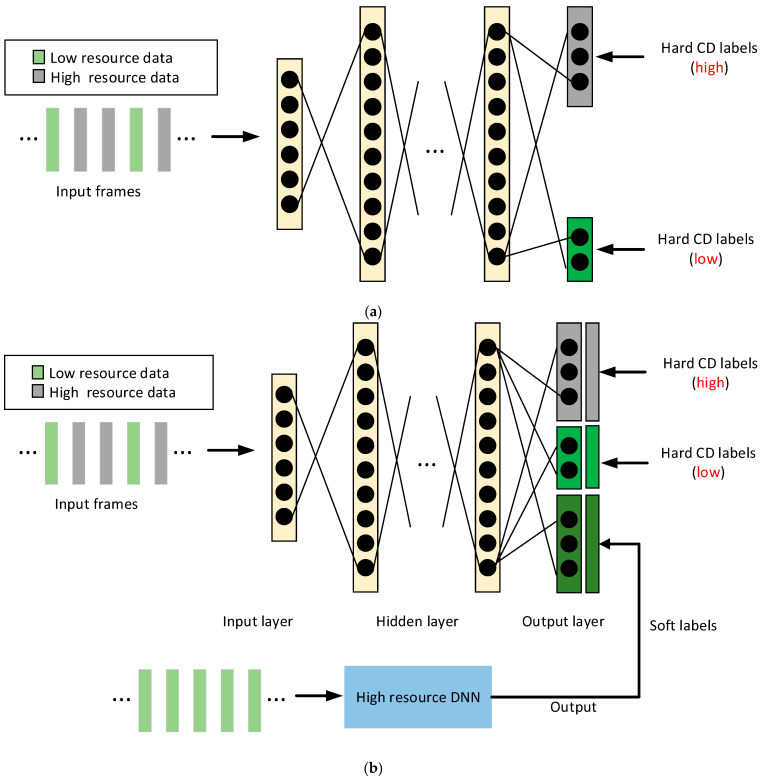
(**a**). The blocksoftmax framework [77]; (**b**). The proposed model framework [77].

**Figure 13 sensors-23-09096-f013:**
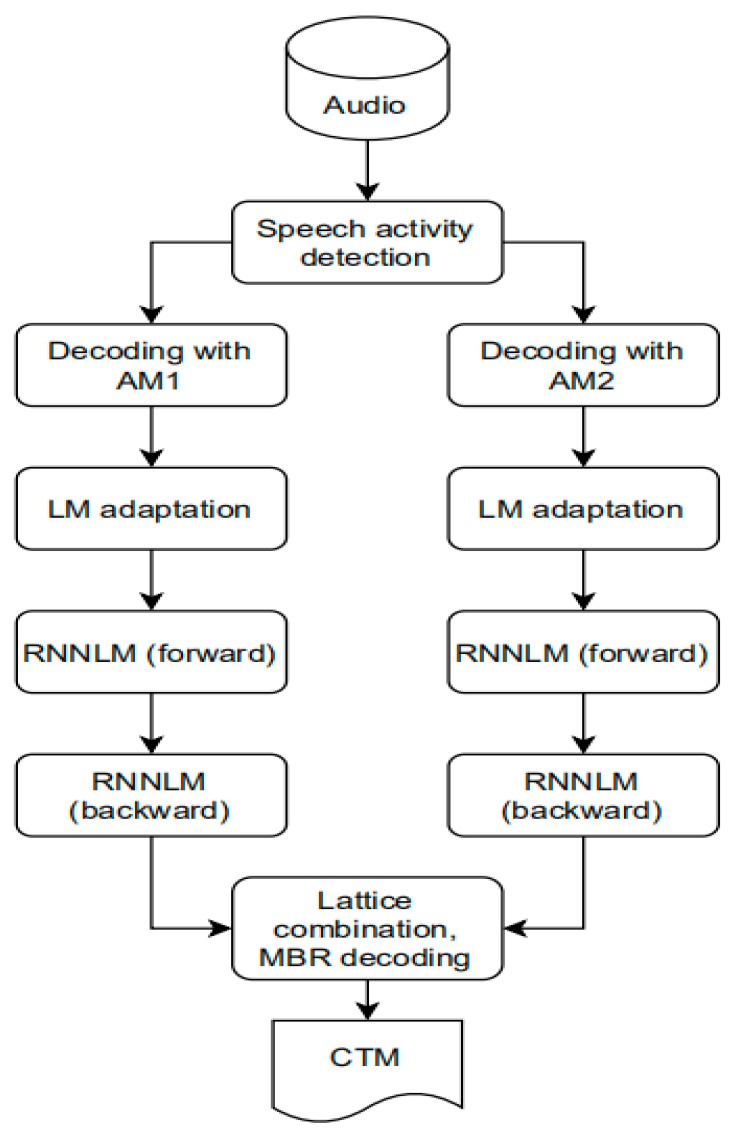
Workflow of the decoding pipeline [79].

**Figure 14 sensors-23-09096-f014:**
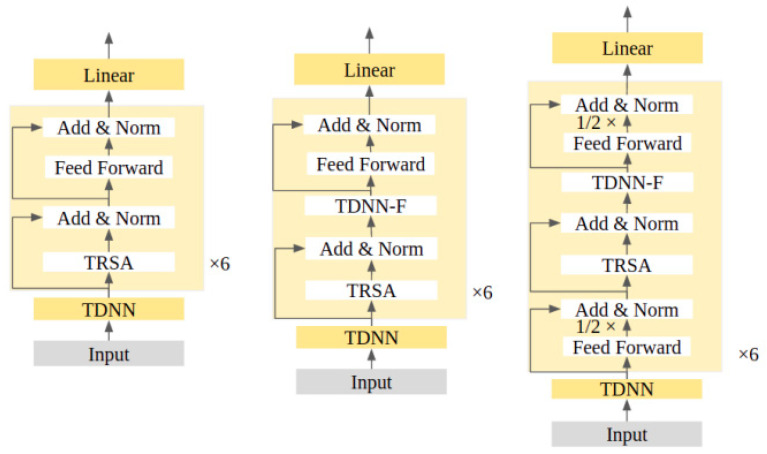
Different acoustic models for our DNN-HMM systems. From left to right are TRSA-Transformer; TRSA-Transformer + TDNN-F and TRSA-Transformer + TDNN-F + Macaron-FFN [80].

**Figure 15 sensors-23-09096-f015:**
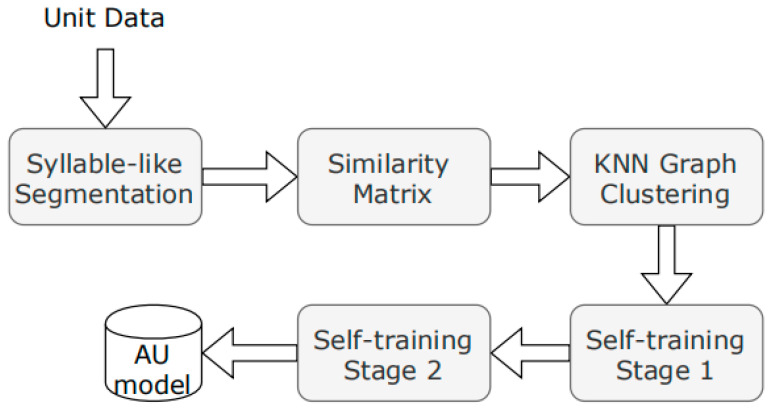
Discovery Acoustic Unit (DAU) Process [114].

**Figure 16 sensors-23-09096-f016:**
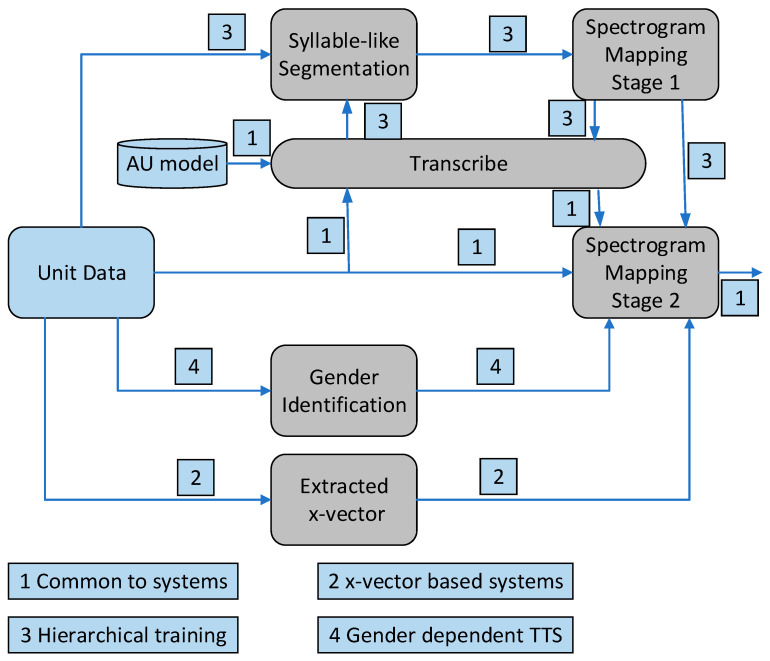
The four proposed synthesis systems [114].

**Figure 17 sensors-23-09096-f017:**
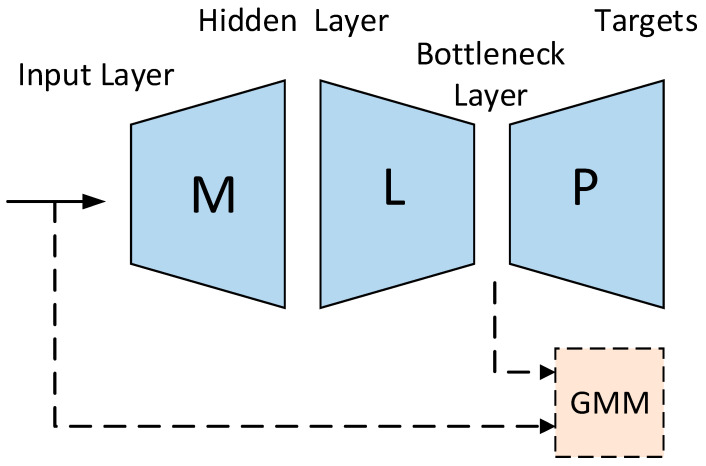
Schematic diagram of tandem method [117].

**Figure 18 sensors-23-09096-f018:**
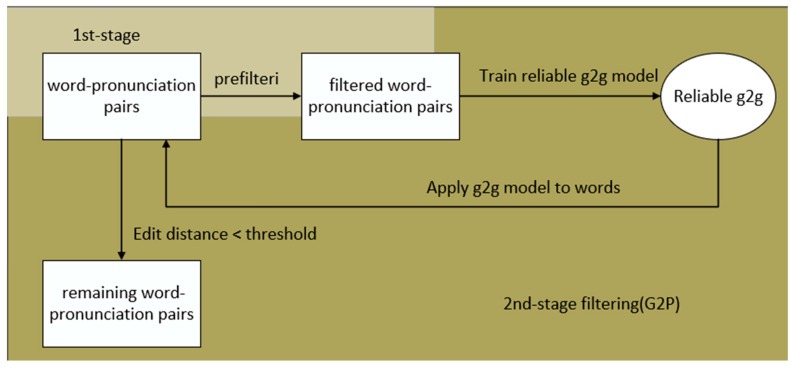
Schematic diagram of two-stage filtration.

**Table 1 sensors-23-09096-t001:** Feature transformation methods.

Method	Database	Principle	Improve Characteristics	Ref
LDA	ARPA RM1	Statistical pattern classification technology, through linear transformation to improve the resolution of feature vectors and compress the information content related to classification	Discrimination and dimensionality reduction	[22]
	TI/NBS connected digit database			[23]
	A 30-word single syllable highly confusable vocabulary			[24]
	CVC syllables database			[25]
HAD	TI-DIGITS	Promote LDA to deal with heteroscedasticity	Discrimination and dimensionality reduction	[26]
GLRDA	200 h of MATBN Mandarin television news	Find a low-dimensional feature subspace by avoiding the most chaotic situation described by the null hypothesis as possible	Discrimination and dimensionality reduction	[27]
MLLR	ARPA RM1	Use a set of regression-based transformations to adjust the mean parameters of HMM	Robustness	[28]
FMLLR	A broadcast transcription task	Constrained MLLR, that is, the mean and variance of Gaussian share a linear transformation matrix	Robustness	[29]
	English in the Callhome database			[30]
VLTN	Two telephone-based, connected digital databases	Appropriate spectral distortion is performed on the frequency spectrum of the speech frame to reduce the spectral variation between different speakers	Robustness	[31]
	Wall Street Journal			[32]

**Table 2 sensors-23-09096-t002:** Extract deep features.

Model	Feature	Method to Realize	Database	Ref
MLP	Tandem feature	Deep feature	Aurora multi-condition database	[37]
MLP	BN feature	Deep feature	Complete NIST, ISL, AMI, and ICSI conference data composition	[38]
MLP	BN feature	Deep feature and multilingual corpus assistance	GlobalPhone database	[39]
DNN	BN feature	Deep feature	70 h of Mandarin transcription task and 309 h of Switchboard task	[40]
MLP	Posterior feature	High-resource corpus assistance	OGI-Stories and OGI-Numbers	[41]
MLP	Posterior feature	High-resource corpus assistance	DARPA GALE Mandarin database	[42]
MLP	Posterior feature	High-resource corpus assistance	English, Mandarin, and Mediterranean Arabic	[43]
DNN	-	Multilingual corpus assistance	138 h in French, 195 h in German, 63 h in Spanish, and 63 h in Italian	[44]
DNN	Mel filterbank, three different pitch features, and fundamental frequency variance	Feature stitching	Tamil	[45]
MLP	The BN features corresponding to the long and short-term complementary features separately trained with MLP	Feature stitching	English, German, and Spanish in the Callhome database	[46]
MLP	High-resource posterior features and low-resource acoustic features	High-resource corpus assistance and feature stitching	English, Spanish, and German in the Callhome database	[47]
DNN	Extract BN features from DNN weight decomposition matrix	Deep feature	Turkish, Assamese, and Bengali in the Babel project	[48]
CNN	Convolutional network neurons	Multilingual corpus assistance	Tagalog in the Babel project	[49]
standard and data-driven auditory filterbanks	AM feature	-	Gujarati	[50]
adversarial end-to-end acoustic model	Language-independent BN feature	Multilingual corpus assistance	Assamese, Bengali, Kurmanji, Lithuanian, Pashto, Turkish, and Vietnamese in the IARPA Babel	[51]
BLSTM & TDNN	AF and SBF	Multilingual corpus assistance and feature stitching	Gujarati, Tamil, and Telugu	[52]

**Table 3 sensors-23-09096-t003:** Acoustic model.

Year	Database	Acoustic Model/Main Content	Method to Realize	Ref
2012	TIMIT	DNN-HMM	-	[67]
2012	Business search	CD-DNN-HMM	-	[68]
2012	Greek speech hdat (II)	KL-HMM	Borrow data	[69]
2011	HIWIRE	Random phoneme space conversion	Borrow data	[70]
2012	Callhome	The mapping idea solves the mismatch of the posterior probabilities of the phonemes of different language modeling units during multilingual training	Borrow data	[71]
2015	Babel Bengali	CNN-HMM	-	[72]
2018	Javanese and Amharic in IARPA Babel	multilingual BLSTM	Borrow data	[73]
2013	French, German, Spanish, Italian, English, and Chinese	SHL-MDNN	Borrow data	[44]
2017	Callhome	SHL-MSLTM	Borrow data	[74]
2018	Tamil, Telugu, and Gujarati	multilingual TDNN	Borrow data	[75]
2018	Tamil, Telugu, and Gujarati	DTNN low-rank DTNN low-rank TDNN with transfer learning	Borrow data	[76]
2017	Hindi, Tamil, and Kannada in MANDI	KLD-MDNN	Borrow data	[77]
2021	Cantonese and Mongolian	CNN-TDNN-F-A	-	[78]
2021	Restricted training conditions for all 10 languages of OpenASR20	CNN-TDNN-F	Borrow data	[79]
2021	Taiwanese Hokkien	TRSA-Transformer TRSA-Transformer + TDNN-F TRSA-Transformer + TDNN-F + Macaron	-	[80]
2021	MATERIAL	CNN-TDNN-F	Borrow data	[81]
2021	Seneca, Wolof, Amharic, Iban, and Bemba	SGMM DNN WireNet	Compression model and Borrow data	[82]
2013	Wall Street Journal	Regularize GMM	Compression model	[83]
2010	Callhome	SGMM	Compression model	[84]
2010	Callhome	Method of multilingual training SGMM	Compression model and Borrow data	[85]
2011	Callhome	Data borrowing method combined with SGMM	Compression model and Borrow data	[86]
2013	GlobalPhone	SMM	Compression model and Borrow data	[87]

**Table 4 sensors-23-09096-t004:** Model parameters and their quantities of GMM.

Parameter Name	Number of Parameters
Gaussian Mean	D × J × I
Covariance Matrix	D × J × I
Gaussian Weight	J × I

**Table 5 sensors-23-09096-t005:** Common activation functions in DNN.

Activation Function	Mathematical Formula
Sigmoid	$f\left(z\right)=\frac{1}{1+e^{-z}}$
Tanh	$\tanh\left(z\right)=\frac{\sinh\left(z\right)}{\cosh\left(z\right)}=\frac{e^{z}-e^{-z}}{e^{z}+e^{-z}}$
ReLU	$f\left(z\right)=\max\left(0,z\right)$

**Table 6 sensors-23-09096-t006:** Model parameters of DNN and their numbers.

Parameter Name	Number of Parameters
Number of input layers	1
Number of hidden layers	B
Number of output layers	1
Number of input layer nodes	A
Number of hidden layer nodes	B × C
Number of output layer nodes	D × E
Weights between the input layer and the hidden layer	B × C × A
Weights between the hidden layer and the hidden layer	B × C × B × C
Weights between the hidden layer and the output layer	D × E × B × C
Offset between input layer and hidden layer	B × C
Offset between hidden layer and hidden layer	B × C
Offset between hidden layer and output layer	D × E
Hidden layer activation function	≥B

**Table 7 sensors-23-09096-t007:** Resource expansion.

Database	Extended Resources	Existing Resources	Generate Resources	Means of Extension	Ref
TIMIT	Data resource	Text	Speech	VTLP	[108]
Limited data packages for Assamese and Zulu in the IARPA Babel project	Data resource	Text	Speech	VTLP	[109]
Wall Street Journal	Data resource	Text	Speech	Add noise	[110]
Switchboard	Data resource	Text	Speech	Add noise	[111]
Hausa in GlobalPhone and Wolof language collected	Data resource	Text	Speech	SP	[112]
English and Indonesian	Data resource	Without Text	Speech	Text-to-Speech (TTS)	[113]
English and Surprise	Data resource	Without Text	Speech	Text-to-Speech (TTS)	[114]
Vietnamese in the IARPA project	Data resource	Speech	text	Semi-supervised training	[115]
Limited data packages for Assamese and Haitian Creole in the IARPA Babel project	Data resource	Text	Speech	VTLP and SFM	[116]
Limited data packages for Assamese and Zulu in the IARPA Babel project	Data resource	Speech Text	TextSpeech	Semi-supervised training and VTLP	[117]
Word pronunciation in the Wiki dictionary website	Pronunciation dictionary	Text	Speech	Crawl the pronunciation of words in the Wiki dictionary website to construct pronunciation dictionary	[118]
On the database of various English pronunciations	Pronunciation dictionary	Text	Speech	Joint sequence model	[119]
U.S. English CMU	Pronunciation dictionary	Text	Speech	G2P model based on LSTM	[120]
CELEX lexical Database of English version	Pronunciation dictionary	Text	Speech	Use a random G2P model to confirm the error items in the newly generated dictionary	[121]
Wiktionary and GlobalPhone Hausa Dictionary and English-Mandarin Code-Switch Dictionary	Pronunciation dictionary	Text	Speech	Fully automatic pronunciation dictionary error recovery method	[122]
Switchboard	Pronunciation dictionary	Text	Speech	Use acoustic features to generate pronunciation dictionary	[123]

**Table 8 sensors-23-09096-t008:** Use combination.

Acoustic Model	Feature	Feature Transformation	Whether the Feature Is Spliced
GMM-HMM	MFCC	LDA + MLLT	Yes
DNN-HMM	Fbank	-	No

**Table 9 sensors-23-09096-t009:** Classical methods.

Database	Feature	Acoustic Model	Best WER (%) (1)/Improvement (2)	Ref
Aurora multi-condition database	Tandem feature	hybrid connectionist-HMM	35 (1)	[37]
Complete NIST, ISL, AMI, and ICSI conference data composition	BN feature	GMM-HMM	24.9 (1)	[38]
GlobalPhone database	BN feature	Multilingual Artificial Neural Network (ANN)	1–2 (2)	[39]
70 h of Mandarin transcription task and 309 h of Switchboard task	BN feature	GMM-HMM	2–3 (2)	[40]
OGI-Stories and OGI-Numbers	Posterior feature	GMM-HMM	11 (2)	[41]
138 h in French, 195 h in German, 63 h in Spanish, and 63 h in Italian	-	SHL-MDNN	3–5 (2)	[44]
Tamil	Mel filterbank, three different pitch feature, and fundamental frequency variance	DNN-HMM	1.4 (2)	[45]
English, German, and Spanish in the Callhome database	The BN features corresponding to the long and short-term complementary features separately trained with MLP	MLP	30 (2)	[46]
English, Spanish, and German in the Callhome database	High-resource posterior features and low-resource acoustic features	MLP	11 (2)	[47]
Turkish, Assamese, and Bengali in the Babel project	Extract BN features from DNN weight decomposition matrix	Stacked DNN	10.6 (2)	[48]
Tagalog in the Babel project	Convolutional network neurons	CNN-HMM	2.5 (2)	[49]
Gujarati	AM feature	TDNN	1.89 (2)	[50]
Assamese, Bengali, Kurmanji, Lithuanian, Pashto, Turkish, and and Vietnamese in the IARPA Babel	Language-independent BN feature	Adversarial E2E	9.7 (2)	[51]
Gujarati, Tamil, and Telugu	AF and SBF	TDNN	3.4 (2)	[52]
TIMIT	MFCC	DNN-HMM	12.3 (2)	[67]
Business search	MFCC	CD-DNN-HMM	5.8–9.2 (2)	[68]
Greek speech hdat (II)	MFCC-PLP	KL-HMM	77 (1)	[69]
Babel Bengali	Fbank	CNN-HMM	2–3 (2)	[72]
Javanese and Amharic in IARPA Babel	Fbank	multilingual BLSTM	5.5 (2)	[73]
Callhome	PLP	SHL-MSLTM	2.1–6.8 (2)	[74]
Tamil, Telugu, and Gujarati	MFCC	multilingual TDNN	0.97–2.38 (2)	[75]
Tamil, Telugu, and Gujarati	Fbank	DTNNlow-rank DTNNlow-rank TDNN with transfer learning	13.92/14.71/14.06 (1)	[76]
Hindi, Tamil, and Kannada in MANDI	Fbank	KLD-MDNN	1.06–3.64 (2)	[77]
Cantonese and Mongolian	i-vectors and hires-MFCC	CNN-TDNN-F-A	48.3/40.2/52.4/44.9 (1)	[78]
Restricted training conditions for all 10 languages of OpenASR20	Fbank	CNN-TDNNF	5–10 (2)	[79]
Taiwanese Hokkien	Fbank	TRSA-Transformer TRSA-Transformer + TDNN-F TRSA-Transformer + TDNN-F + Macaron	43.4 (1)	[80]
MATERIAL	Fbank	CNN-TDNN-F	7.1 (2)	[81]
Wall Street Journal	MFCC	Regularize GMM	10.9–16.5 (2)	[83]
Callhome	MFCC	Method of multilingual training SGMM	8 (2)	[85]
Callhome	MFCC	Data borrowing method combined with SGMM	1.19–1.72 (2)	[86]
GlobalPhone	MFCC	SMM	1.0–5.2 (2)	[87]

## Data Availability

Not applicable.

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
