# Peer review of "Frontier Research on Low-Resource Speech Recognition Technology"

_sensors, 2023, doi:10.3390/s23229096_

Round 1
Reviewer 1 Report
Comments and Suggestions for Authors
This is an overview article, but the majority of article is from 2010-2014, the latest one is from 2019 and it is the only reference from that year.
Check grammar as some sentences are difficult to understand, e.g. “Analyzed the many technical…”, “Figure 1 is the basic…”,
What is the difference between current speech recognition rate and continuous speech recognition rate? What is meant by current in this sentence?
Reformulate contributions on 2 page, especially 3) one, which is difficult to understand
Speech features should have characteristics: 2) ”… try to eliminate speech Human,…” it does not make sense
Whole section regarding speech features focused on speech recognition and analysis is missing, providing basic classification of many features that exist so far, giving pros and cons of each, e.g. acoustic, prosodic, vocal tract, pre glottal, in time, in frequency, generative, model based, etc.
It is stated that feature transformation methods can be divided into classifier dependent and independent, and where the methods, e.g. [21- 24] etc. belong to? Is there a special category for them?
Occurrences of HAD should be changed to HDA. Moreover, is it a linear transform? If so, then the common term HLDA should be used instead.
Reformulate this: “Experiments have proved that, compared with the case without LDA, when the class of LDA transformation is defined as a subphone unit, the maximum accuracy can be improved, and the error rate is reduced by one-fifth [21]”. It is not clear what is compared to what, with or without LDA.
Put results describing certain group of experiments into table instead of having them in a sentence, so that it is more legible and easy to compare. This applies to several places in the text.
Some classes of techniques, features, methods are not well defined or explained, e.g. posterior features, page 5, etc.
What is the exact difference between MLP an DNN features? MLP can be also a DNN, what is the exact difference.
Some abbreviations are used without prior definition, e.g. BLSTM
Figure 3, it is not clear which part is b and c.
How the items in table 3 are organized? some order/ grouping should be kept, it seems it is rather random.
Unnecessary redundancy- repetition: “Each language has a 50-hour high-resource data set and a 10-hour low-resource data set. The team did three sets of experiments: Each language has a 50-hour high-resource data set and a 10- hour low-resource data set.”
Page 21, µ*δ should be probably µ-δ
Statement: “Table 2 shows that shallow features such as MFCC and PLP are not robust enough.” As no explicit features are mentioned and no results are stated in Tab 2, this cannot be inferred from table 2, it should be added into the table.
No references are given in relation to “some other neural networks have emerged in recent years, such as long and short-term memory loop neural networks, connection timing classification, multi-dimensional RNN, hierarchical down-sampling RNN”.
A discussion section is missing that would summarize the main findings in forms of tables listing methods results, settings, languages etc. This is very important for this survey, e.g. which acoustic features to use, how they should be transformed and when, spliced or fused, what acoustic models to used and when, etc. It is very important to have a separate summarizing/ comparison table for each area the article focuses on, ranking the methods and their setting, etc. Further, discuss if some results can be generalized or some findings apply only to certain languages – cases etc. This is the most important part for anybody who is planning to build such system, i.e. to have some answers to issues that were discussed in the article. Not having such comparison tables and summarizing- generalizing discussion is degrading the contribution of the article.
Very little was said about CTC and transformers?
Author Response
Dear anonymous reviewer 1:
Thank you very much for your affirmation of our article and your valuable comments. The title of our article is “Frontier Research on Low-resource Speech Recognition Technology” (ID: sensors-1556519). Your comments are of great help to the improvement of our thesis, also have important guiding significance for our research. We have carefully studied your comments and made revisions. The main corrections in the paper and the responds to the reviewer’s comments are as flowing:
Point 1: Check grammar as some sentences are difficult to understand, e.g. “Analyzed the many technical…”, “Figure 1 is the basic…”.
Response 1: As you said, there are indeed some sentences in the article that are difficult to understand. Your suggestions are very important for us to improve the grammar of the article. We have checked relevant dictionaries and consulted professional translators to improve the grammar of the article.
Point 2: What is the difference between current speech recognition rate and continuous speech recognition rate? What is meant by current in this sentence?
Response 2: Thanks a lot for your opinion. Because of your comments, our articles will be more complete. Otherwise, unnecessary ambiguity will be created. Below is an edit to the question you asked. Continuous speech recognition is to recognize continuous speech signals into corresponding texts. Usually, speech recognition refers to continuous speech recognition unless otherwise specified. The difference here is the low-resource language recognition rate and the large-vocabulary speech recognition rate. Therefore, the current speech recognition rate in the previous version refers to the low-resource speech recognition rate. This section of the ambiguity has been revised based on the reviewer's suggestion. Lines 28-34 of the article are the modified content.
The specific modifications are as follows. Continuous speech recognition is to recognize continuous speech signals into corresponding texts. Usually, speech recognition refers to continuous speech recognition unless otherwise specified. However, in a low-resource environment, due to the lack of annotated corpus, the model training of the speech recognition system usually occurs overfitting and cannot achieve good recognition performance. As far as current research is concerned, although a large number of targeted studies have made some progress relative to traditional modeling methods, there is still a huge gap between the low-resource speech recognition rate and the large-vocabulary speech recognition rate.
Point 3: Reformulate contributions on 2 page, especially 3) one, which is difficult to understand Speech features should have characteristics: 2)” try to eliminate speech Human, …” it does not make sense Whole section regarding speech features focused on speech recognition and analysis is missing, providing basic classification of many features that exist so far, giving pros and cons of each, e.g. acoustic, prosodic, vocal tract, pre glottal, in time, in frequency, generative, model based, etc.
Response 3: I think it's important to me that your question about whether speech has these few characteristics and your suggestion to rewrite this paragraph. If you hadn't pointed out the potential ambiguity in this place, it would be easy for others to misunderstand this article. This will affect the quality of this article. So, I thank you from the bottom of my heart. Again, your suggestions are really important for us to improve the article. There is a real problem with the language of this paragraph of the article. Therefore, it has been revised according to the comments of the reviewers. Lines 107-113 of the article are the modified content. The specific modifications are as follows. Feature extraction is to turn the speech signal into a feature parameter that the recognizer can process. Only when the feature parameter has the following three characteristics can a good speech recognition rate be obtained. 1) Distinction, which can accurately model speech acoustic units; 2) Robustness, in the face of changes in speakers and channels, feature parameters will not be affected, and feature parameters can resist noise interference; 3) Low dimensionality, while containing enough effective information, the feature dimension should be as low as possible to reduce the data scale and improve the system efficiency. We would also like to describe what the reviewer said "difficult to understand speech has these characteristics and to describe the advantages and disadvantages of speech characteristics such as acoustics, prosody, vocal tract, preglottic, time, frequency, generative, model-based, etc."Explain. Not-speech features have the following three characteristics: discriminative, robust and low-dimensional. Feature extraction is to turn the speech signal into feature parameters that the recognizer can process. It is that the feature parameters after feature extraction have three characteristics: discrimination, robustness and low dimensionality. Only when the feature parameters have these three characteristics can a good speech recognition rate be obtained. In addition, the reason why features such as acoustics, prosody, vocal tract, preglottic, time, frequency, generative, model-based, etc. are not given here is because the features to be described in this small part of feature extraction are not the features of speech, it is the characteristics of the feature parameters after the feature extraction operation.
Point 4: It is stated that feature transformation methods can be divided into classifier dependent and independent, and where the methods, e.g. [21-24] etc. belong to? Is there a special category for them?
Response 4: Regarding the revision suggestion given by the reviewer for the "feature transformation method can be divided into classifier-dependent and independent." In order to avoid cognitive errors and have no impact on the following article, remove this sentence.
Point 5: Occurrences of HAD should be changed to HDA. Moreover, is it a linear transform? If so, then the common term HLDA should be used instead. Reformulate this: “Experiments have proved that, compared with the case without LDA, when the class of LDA transformation is defined as a subphone unit, the maximum accuracy can be improved, and the error rate is reduced by one-fifth [21]”. It is not clear what is compared to what, with or without LDA.
Response 5: Sorry for such a spelling error. At the same time, I am also very grateful to the reviewers for their serious, responsible and rigorous attitude. It is because of your seriousness that we have avoided such a low-level spelling error. Your opinion is very important to the improvement of our article. There is a real problem with the language of this paragraph of the article. Therefore, it has been revised according to the comments of the reviewers. The HAD that the reviewer said has been changed to HDA. The modified content is on line 146 of the article. It has been restated: "Experiments show that the maximum accuracy can be improved and the error rate reduced by one-fifth when the class of LDA transformation is defined as a consonant unit compared with the case without LDA [21]". Lines 136-138 of the article are the modified content.
The specific modifications are as follows. Experiments show that using the LDA acoustic feature transformation method reduces the recognition error rate by one-fifth compared to the case without using the LDA acoustic feature transformation method [21]. There is LDA here. Here, the case of not using the LDA acoustic feature transformation method is compared with the LDA acoustic feature transformation method. HAD is a linear transformation, a generalization of LDA. The "H" in HAD is short for heteroscedasticity. And LDA is used for homoscedasticity. Therefore, although HAD is a linear transformation, it is not expressed by HLDA because of its different application scope from LDA.
Point 6: What is the exact difference between MLP an DNN features? MLP can be also a DNN, what is the exact difference.
Response 6: Your suggestion of connection and distinction between MLPs and DNNs makes perfect sense. We can have a deeper understanding of the nature of these two. We have supplemented this connection and difference between MLP and DNN by consulting the literature. Lines 182-190 of the article are supplementary content. The specific contents are as follows. DNN is also an MLP in essence. However, due to the limitation of theoretical research level and hardware computing power in the past, the MLP used by researchers usually has a small number of hidden layers (1~2), and the network parameters are randomly initialized. Therefore, MLP is now generally used to refer to such a Neural Networks. The DNN is expanded on the original MLP structure, and its hidden layers become more (generally more than 5), and the category target granularity of the output layer is smaller than the original, and the number of categories is more. Moreover, it usually uses some effective pre-training methods to replace the initialization of the original random network parameters, which effectively guarantees the speed and accuracy of DNN training.
Point 7: Some abbreviations are used without prior definition, e.g. BLSTM Figure 3, it is not clear which part is b and c.
Response 7: We appreciate your suggestions. At the same time, in the team, repeated deliberation and consideration were carried out. It must be said that your suggestions played a vital role in the quality of our articles. But here we need some explanation. The following is an interpretation of Figure 3 proposed by the reviewers. In this place, the abbreviations are defined in advance. There are mainly three abbreviations involved here: Articulatory Feature (AF), Stacked Bottleneck Feature (SBF) and Bidirectional Long Short-Term Memory (BLSTM) network. Figure 3(b) is the pronunciation classifier used to extract AF. Because building an efficient pronunciation classifier requires a lot of data, so we trained the BLSTM pronunciation classifier by pooling data from available low-resource Indian languages (Gujarati, Tamil and Telugu). Figure 3(c) is the feature extractor used to extract SBF.
Point 8: How the items in table 3 are organized? some order/ grouping should be kept, it seems it is rather random.
Response 8: For the reviewer's question, we have a serious discussion and research. On behalf of the entire team, I extend my deepest respect and gratitude to the reviewers. Thank you very much for your valuable comments. But here we need some explanation. Explain the doubts raised by the reviewers about the components of Table 3. The specific explanation is as follows. Table 3 is composed of five parts, the specific parts are as follows: year, data set used in experiment, acoustic model used in experiment or key problems solved by modeling for acoustic model, means to build a good acoustic model and experimental references. The third column is the acoustic model used in the experiment or the key problem solved by the modeling of the acoustic model. In column 3, except for the two lines corresponding to references [72][73], the rest are the names of the acoustic models used in the experiments. References [72][73] are the key problems for the modeling of acoustic models. Among them, the literature [72] is a solution to the problems existing in phoneme modeling. The literature [73] is a solution to the problem of phoneme posterior probability mismatch between different language modeling units during multilingual training. Column 4 is the means by which a good acoustic model is built. There are two main types, as follows. One is to perform auxiliary training by introducing data from other languages to the target acoustic model, that is, borrowing data from languages with rich resources to improve the training effect of the target acoustic model, and ultimately improve the recognition rate of low-resource speech recognition systems. The other is to compress the parameters of the acoustic model, that is, to reduce the model parameters. The SGMM model is a typical compression model. Columns 1, 2, and 5 need no explanation, just plain and simple, just the year, the dataset, and the reference. Therefore, the combination of Table 3 is not arbitrary.
Point 9: Unnecessary redundancy- repetition: “Each language has a 50-hour high-resource data set and a 10-hour low-resource data set. The team did three sets of experiments: Each language has a 50-hour high-resource data set and a 10- hour low-resource data set.”
Response 9: Immediately after the reviewer pointed out redundancy in this paragraph of the article, we checked it and found that, in this place, there is indeed unnecessary repetition. Really appreciate the reviewer's word-for-word checking attitude to avoid these problems. According to the reviewer's suggestion, removing these redundancies does make the article more streamlined. “Each language has a 50-hour high-resource dataset and a 10-hour low-resource dataset. The team conducted three sets of experiments: each language had a 50-hour high-resource dataset and a 10-hour low-resource dataset." This paragraph contains unnecessary repetitions raised by the reviewers, so the redundancies are removed as requested. Lines 488-495 of the article are the modified content. The specific contents are as follows. Each language has a 50-hour high-resource data set and a 10-hour low-resource data set. The team conducted three comparative experiments: 1) a DNN baseline system using 10 hours of monolingual language; 2) Blocksoftmax system using 50-hour high-resource data sets in both languages and 10-hour low-resource data sets in test languages; 3) Blocksoftmax system with additional KLD using 50-hour high resource data sets in both languages and 10-hour low resource data sets in test languages.
Point 10: Page 21, µ*δ should be probably µ-δ?
Response 10: Explain the question raised by the reviewer that µ*δ may be µ-δ on page 21. It is indeed µ*δ not µ-δ here. Thank reviewers for your consideration and serious attitude towards this passage. Thanks again for your suggestion.
Point 11: Statement: “Table 2 shows that shallow features such as MFCC and PLP are not robust enough.” As no explicit features are mentioned and no results are stated in Tab 2, this cannot be inferred from table 2, it should be added into the table.
Response 11: We have thought carefully about your suggestion and tried our best to revise it. Your opinion is very important to the improvement of our article. After discussion, we have revised the reviewer's suggestion. "Table 2 shows that shallow features such as MFCC and PLP are not robust enough." Since explicit features are not mentioned in Table 2 and the results are not stated, it cannot be inferred from Table 2 and should be added to the table. For the reviewer's suggestion to revise this sentence, I think it has important reference significance. But because my summary in this part is to highlight the importance of extracting deep features, rather than talking about the two common traditional shallow features of MFCC and PLP. So it's good to pass the MFCC and PLP in this part. So here I deleted "as can be seen from Table 2" and rephrased the original sentence. Lines 738-739 of the article are the modified content. The specific contents are as follows. In terms of feature extraction, common traditional shallow features such as MFCC and PLP are not robust enough.
Point 12: No references are given in relation to “some other neural networks have emerged in recent years, such as long and short-term memory loop neural networks, connection timing classification, multi-dimensional RNN, hierarchical down-sampling RNN”.
Response 12: Many thanks to the reviewers for pointing out possible issues here. But our focus may conflict with your understanding of this place. For the reviewer's comment on "some other neural networks that have appeared in recent years, such as long short-term memory recurrent neural network, connection time series classification, multi-dimensional RNN, hierarchical downsampling RNN, did not give any reference", make an explanation. The article covers long short-term memory recurrent neural networks and connection timing classification. The BLSTM mentioned many times in the article is an extension of LSTM and is a bidirectional LSTM. And MLSTM is a multilingual LSTM. For multi-dimensional RNNs and hierarchical downsampling RNNs, I think it is better to leave them in the technical challenges and prospects section at the end. The multi-dimensional RNN and the hierarchical down-sampling RNN will not be described in detail, as long as it is known that these two networks can make full use of the network structure of the rich information in the speech signal and effectively improve the performance of low-resource speech recognition. This also serves the purpose of writing this paragraph at the conclusion.
Point 13: Very little was said about CTC and transformers?
Response 13: For the reviewer's question as to why this part of the content is less involved, I will explain our intentions to the reviewer here. Because CTC and transformers are in the end-to-end domain, there is little to talk about. Although end-to-end speech recognition has appeared in recent years, it is not really suitable for low-resource speech recognition because of the large amount of speech data required to train its own network. We can only hope that in future research, end-to-end techniques can be truly applied to low-resource speech recognition. Provide a new idea for our low-resource speech recognition.
Special thanks to you for your good comments. We tried our best to improve the manuscript and made some changes in the manuscript. These changes will not influence the content and framework of the paper. And here we did not list the changes but marked in yellow in revised paper. We appreciate for Editors/Reviewers’ warm work earnestly, and hope that the correction will meet with approval.
Once again, thank you very much for your comments and suggestions.
With best regards,
Yours sincerely,
All authors.

Reviewer 2 Report
Comments and Suggestions for Authors
This paper describes a survey for low-research speech recognition. Although it is valuable , it is incomplete.
1. Please introduce the history of this research field such as
/ Workshop on spoken language technologies for under-resource languages, 2008-2021
/Interspeech 2018: Special session, low resource speech recognition challenge for Indian languages
/Interspeech 2020: Special session, zero-resource speech challenge
/interspeech 2021: Special session, Open ASR20 and low-resource ASR development
2. This survey focuses on studies from 2012 to 2018, but not recent studies from 2019 to 2021. Therefore, it describes GMM-HMM approaches more than half. Please classify each section into subsections of (a)general approach, (b) GMM-HMM approach and (c) DNN-based approach.
3. There is no description of robust feature extraction from raw speech waves based on CNN.
4. There is no description for data-augmentation based on text-to speech.
5. Is "transfer learning" in page 14 the same as "teacher-student model" ?
6. Although many improvement results are introduced, these improvements do not show significant differences statistically. In general, error reduction rate more than 10% is necessary to show the effectiveness for useful approaches.
7. "SGMM" and "SHL-MDNN" are described at plural sections.
8. Please compare the number of parameters between typical GMM-HMM approaches and typical DNN-based approaches in Section 4.
Author Response
Response to Reviewer 2 Comments
Dear anonymous reviewer 2:
Thank you very much for your affirmation of our article and your valuable comments. The title of our article is “Frontier Research on Low-resource Speech Recognition Technology” (ID: sensors-1556519). We are trying our best to make it a satisfactory article. There is no doubt that your comments are of great help to the improvement of our thesis, also have important guiding significance for our research. We have studied your suggestions carefully and made some modifications. The main modifications and responses to reviewers' comments in the paper are as follows:
Point 1: This survey focuses on studies from 2012 to 2018, but not recent studies from 2019 to 2021. Therefore, it describes GMM-HMM approaches more than half. Please classify each section into subsections of (a)general approach, (b) GMM-HMM approach and (c) DNN-based approach.
Response 1: First of all, I greatly appreciate the reviewer's suggestion to break each section into three subsections. But for this question, we need some explanation. Regarding the reviewers' comments on "This survey describes nearly half of GMM-HMM. Please classify each section as a subsection of (a) general methods, (b) GMM-HMM methods and (c) DNA-based methods" suggested explanation. In the article, although GMM-HMM has been appearing, it is said that DNN-related methods are introduced because of its shortcomings. The focus has always been related to deep learning, not GMM-HMM, so there is no need to classify it. The role of GMM-HMM is an introduction.
Point 2: There is no description of robust feature extraction from raw speech waves based on CNN.
Response 2: For the reviewer who said that there is no question about extracting deep features with CNN in the article, I will explain. Of course, many thanks to the reviewers for their suggestions. Regarding the reviewer's explanation for "The work has no description of robust CNN-based feature extraction". The article describes the use of CNNs to extract robust deep features. In [50], the multilingual training of CNN exploits the idea of parameter sharing, where convolutional layers and fully connected layers are shared on the source language. In other words, the two convolutional stages are used as a general language feature extractor to extract deep features. Lines 314-321 of the article are the content of using CNN to extract deep features. The specific contents are as follows. In [50], the multilingual training of CNN also uses the idea of parameter sharing, sharing the convolutional layer and the fully connected layer on the source language. In other words, these two convolution stages are used as general language feature extractors to extract features from the second max-pooling layer, or from the lowest fully connected layer located above the convolution stage. Compared with DNN, CNN generates better feature representations in cross-language hybrid systems.
Point 3: There is no description for data-augmentation based on text-to-speech.
Response 3: I would like to explain to the reviewer that "there is no text-to-speech approach to augment the data in this article". Of course, I am very grateful to the reviewers for their serious and responsible work. An explanation of the revision suggested by the reviewer for "Data Augmented Description without Text-to-Speech" is as follows. Section 5.1 for extended research on data resources. The expansion of data resources is divided into two parts, one part is to generate new voices through existing texts to achieve the purpose of data enhancement, and the other part is to generate new texts through existing voices to achieve the purpose of data enhancement. Text-to-speech based data augmentation is the first part, generating new speech from existing text. Specific methods include Vocal Tract Length Perturbation (VTLP), adding noise, speed perturbation (SP) and so on. These methods are described in detail in the article. Lines 587-604 of the article describe these methods in detail. The specific contents are as follows. Among them, the corresponding audio generated by the existing text is mainly embodied on the basis of the existing training data, keeping the semantics unchanged (that is, the labeled text unchanged) to transform or process the audio, and change its channel, speech rate and other characteristics. At present, the commonly used methods in research include Vocal Tract Length Perturbation (VTLP), noise addition, and Speed Perturbation (SP). Specifically, VTLP generates a data enhancement scheme for new samples by perturbing or distorting the speech spectrum of existing training samples, linearly stretching or compressing the speech signal in the frequency domain, and changing the speaker's vocal tract length to achieve data enhancement [96, 97]. Adding noise is one of the classic data expansion methods. The purpose of data enhancement is achieved by adding different types of noise with different signal-to-noise ratios to the speech [98, 99]. SP is to generate a regular time signal, which essentially stretches or compresses the speech signal linearly in the time domain. In simple terms, the speaker’s speech rate changes in the same sentence to achieve data enhancement [100].
Point 4: Is "transfer learning" in page 14 the same as "teacher-student model"?
Response 4: Thanks to the reviewers for bringing this question to me. But for this question, we need some explanation. Explain the reviewer's question about whether "transfer learning" on page 14 is the same as the "teacher-student model". Here "transfer learning" is not the same as "teacher-student model".
Point 5: Although many improvement results are introduced, these improvements do not show significant differences statistically. In general, error reduction rate more than 10% is necessary to show the effectiveness for useful approaches.
Response 5: Many thanks to the reviewers for showing me this idea. It is undeniable that the opinions of the reviewers are really very important. But for this question, we need some explanation. For the reviewer's response to the sentence "Despite the many improved results, these improvements were not statistically significantly different. In general, an error reduction rate of more than 10% is necessary to show the effectiveness of a useful method" explain. Admittedly, the reviewer's suggestion is excellent. But in scientific experiments, even if there is a 1% improvement, we have to admit it is an improvement. Because doing scientific research itself is from a little accumulation.
Point 6: "SGMM" and "SHL-MDNN" are described at plural sections.
Response 6: We have taken this reviewer's suggestion into consideration. And our entire team has given thoughtful consideration to the suggestions made by the reviewers. Of course, many thanks to the reviewers for their comments. But for this question, we need some explanation. Explanation for the reviewer's suggestion that "SGMM and SHL-MDNN are described in multiple sections". For SGMM, because it reduces the model size through parameter sharing, it can reduce the amount of data during training. And this is the most important point of SGMM, because this article is not about speech recognition with large amounts of data, but speech recognition with low resources. Moreover, SGMM can also borrow data from other resource-rich languages through multilingual training to achieve the purpose of training its own model. And SMM is also an extension of SGMM. In low-resource research, researchers have done a lot of optimization through the SGMM model. So it deserves a place in the article and is mentioned in various places. The SHL-MDNN model is mentioned twice, but from different perspectives. In the part of extracting deep features, it is used as a general feature extractor. In the case of acoustic modeling, it is used as a basic model that is fully trained by resource-rich multi-languages to transfer low-resource languages. So although they appear twice, they do not conflict.
Point 7: Please compare the number of parameters between typical GMM-HMM approaches and typical DNN-based approaches in Section 4.
Response 7: Your opinion is very important to the improvement of our article. But for this question, we need some explanation. Regarding the reviewer's explanation for "In Section 4, please compare the number of parameters of a typical GMM-HMM method and a typical DNA-based method". This suggestion makes a lot of sense, but we can't find a parameter comparison by looking up sources. Currently there are schematic diagrams of these two methods.
Special thanks to you for your good comments. We tried our best to improve the manuscript and made some changes in the manuscript. These changes will not influence the content and framework of the paper. And here we did not list the changes but marked in yellow in revised paper. We appreciate for Editors/Reviewers’ warm work earnestly, and hope that the correction will meet with approval.
Once again, thank you very much for your comments and suggestions.
With best regards,
Yours sincerely,
All authors.
Round 2
Reviewer 1 Report
Comments and Suggestions for Authors
Some questions were fully and some only partly responded to.
-First of the 3 contributions (page 2) states: “The research status of low-resource speech recognition is described from three aspects: feature extraction, acoustic model and resource expansion.“ However, there are very little said about speech feature extraction itself in section 3. Thus it should be changed to something more related to the section 3.
-The wording of the second contribution sound strange, i.e. “ 2) Analyzed the many technical challenges faced by realizing reliable low-resource language recognition”
- The main issue/ comment was neither incorporated in the article nor commented in the response section. So it still remains open, i.e.:
A discussion section is missing that would summarize the main findings in forms of tables listing methods results, settings, languages etc. This is very important for this survey, e.g. which acoustic features to use, how they should be transformed and when, spliced or fused, what acoustic models to used and when, etc. It is very important to have a separate summarizing/ comparison table for each area the article focuses on, ranking the methods and their setting, etc. Further, discuss if some results can be generalized or some findings apply only to certain languages – cases etc. This is the most important part for anybody who is planning to build such system, i.e. to have some answers to issues that were discussed in the article. Not having such comparison tables and summarizing- generalizing discussion is degrading the contribution of the article.
Author Response
Thanks for your opinion.

Reviewer 2 Report
Comments and Suggestions for Authors
The manuscript was not revised in according with my comments, especially, "This survey focusses on studies from 2012 to 2018, but not recent studies from 2019 to 2021." This survey may miss-lead for readers.
Author Response
Thanks for your opinion.

Round 3
Reviewer 1 Report
Comments and Suggestions for Authors
The authors have extended the article and addresses most of the comments.
Some information now located in discussion section regarding feature extraction should be rather moved to section 3. Moreover try to avoid sentences like: “But next, I summarize which...”, or “Now, I explain the contents of...”. Passive tense would sound more professional.
It is quite misleading to use the term scale in table 4 (number of parameters...).
“...proposed to use CNN to extract thousands of dimensional convolutional network neurons as acoustic features” – this is not clear, thousand neurons, dimensions or features - reformulate this.
General evaluation: Even though a lot of work was indisputably done and lot of methods mentioned the required information that is to be provided by a good survey article is not easily and compactly accessible, i.e. in the forms of comparable tables covering all main aspect the article focuses on, e.g. giving settings and achieved results in a sorted order so that it is clear which method/s to use and when in particular in combination with adjacent processing steps, and why, etc. Some of that information is scattered in the text that is difficult to be found and compared. Finally, more generalizing discussion and reasoning would be beneficial.
Author Response
Thanks for your advice. Your suggestion allows us to complete this paper better.

Reviewer 2 Report
Comments and Suggestions for Authors
The revised paper covers almost approaches for low resource ASR techniques and valuable.
However, there are still some revised points as follows:
- Especially, the survey of recent studies is the explanation of individual systems. You had better categorize such systems into several approaches/techniques for improving readability/understandability.
- You had better describe number of parameters of typical or standard structures in Table 5. (for example, 1x10^7 for L=7, W=DxD, D=2048, ・・・、#states=2000, #mixtures=64,・・・)
- I -> we
Author Response

(The authors gave the same response as above.)
